



# Iberia01: A new gridded dataset of daily precipitation and temperatures over Iberia

Sixto Herrera[a], Rita M. Cardoso[b], Pedro M.M. Soares[b], Fátima Espírito–Santo[c], Pedro Viterbo[c], and José M. Gutiérrez[d]

[a]Meteorology Group. Dept. of Applied Mathematics and Computer Science. Universidad de Cantabria. Santander, Spain
[b]Instituto Dom Luiz (IDL), Facultade de Ciências, Universidade de Lisboa, Lisboa, Portugal
[c]Instituto Português do Mar e da Atmosfera (IPMA), Lisboa, Portugal
[d]Meteorology Group. Instituto de Física de Cantabria, CSIC-University of Cantabria, Santander, Spain

**Correspondence:** Sixto Herrera (sixto.herrera@unican.es)

**Abstract.** The present work introduces a new observational gridded dataset produced using a dense network (thousands) of stations over the Iberian Peninsula (referred to as Iberia01, Gutiérrez et al. (2019), DOI: http://dx.doi.org/10.20350/digitalCSIC/8641), providing daily precipitation and temperatures for the period 1971-2015 at $0.1°$ regular (and $0.11°$ rotated CORDEX compliant) resolutions. A comparison with both the standard and ensemble version of the E-OBS v17 dataset (at $0.25°$ and $0.1°$ resolutions, respectively) is undertaken in order to assess observational uncertainty in this region. First, a standard com-

parison is performed for several weather indices, obtaining the differences between both datasets. Secondly, a new probabilistic intercomparison analysis is introduced, using the E-OBS ensemble (v17e) to characterize observational uncertainty and testing the hypothesis that Iberia01 is a realization of the ensemble (i.e. it falls within the observational uncertainty range provided by E-OBS). Finally, the effective resolution of the auxiliar very high resolution grid ($0.01°$) built to obtain the area-average repre-

sentativity of the final dataset, and thus the possibility to increase the resolution of the dataset by means of pure interpolation methods, is analyzed considering an extreme event of convective precipitation affecting the Iberian Peninsula.

As a result, we show that Iberia01 produces more realistic patterns than E-OBS v17 in the case of precipitation for all the indices considered, although both are comparable for temperatures. These differences are assessed using the probabilistic approach based on the E-OBS ensemble showing a quite homogeneous spatial pattern for precipitation (with less than 25%

significantly —at a 10% level— different days between both datasets) and a very inhomogeneous pattern for temperatures, with either a small (in most of the regions) or large fraction of significantly different days. The great uncertainty of the precipitation given by E-OBS ensemble, in which the standard deviation of the ensemble has the same order than the mean value, increases the significance of the results obtained for this variable reflecting the differences between both datasets.

KEY WORDS: *Observational uncertainty; E-OBS; ensemble; gridded observations; kriging; thin plate splines; extremes; Clime; Precip-

itation; Temperature*



# 1 Introduction

The availability of high resolution climate data together with an estimate of its uncertainty (observational uncertainty) is of paramount importance for climate studies, from global (Sun Qiaohong et al., 2018) to regional and local scales (Kidd et al., 2011). The first comprehensive gridded temperature dataset was obtained by Jones et al. (1982). This dataset only covered the Northern Hemisphere and produced monthly

means at $5°$ latitude by $10°$ longitude grid. Later, this grid was extended to cover the entire globe, with a higher longitudinal resolution (at $0.5°$ resolution, Jones et al., 1986a, b) and currently expands several variables covering Earth's land areas for 1901-2015 (CRU TS4.0, Harris et al., 2014; Trenberth et al., 2014). However, this kind of resolution is too coarse for regional analysis, which typically requires datasets with tens of kilometres spatial resolution and daily to sub-daily temporal data, in order to differentiate climatic sub-regions and extreme events. In Europe, within the framework of the ENSEMBLES project, the first high resolution continental observational gridded

dataset was produced (E-OBS) for daily maximum, minimum and mean temperatures, precipitation (Klein Tank et al., 2002; Haylock et al., 2008; Klok and Klein Tank, 2009) and sea level pressure (van den Besselaar et al., 2011). This is by far the most used climate reference for European climate studies. Yet, in some regions, E-OBS relies on a sparse observational network which limits its ability to correctly represent not only mean values, but also the variance and extremes, particularly over complex topography (Klok and Klein Tank, 2009). The influence of the station density in the quality of gridded products have been analyzed in the last decades by several authors: Rudolf et al. (1994) was

able to significantly reduce the precipitation error, from a maximum of $40\%$ to $20\%$, by doubling the number of stations within a $2.5°$ grid box; Prein and Gobiet (2017) found that in regions with sparse data the uncertainties associated to mean seasonal precipitation could reach $60\%$; Beguería et al. (2016) found that, in a high resolution observationally based gridded dataset, the density of the underlying observations determines its spatial variance and thus strongly influences climate variability; Hofstra et al. (2010) concluded that, by randomly changing the number of stations in each grid box, a reduction in the density of stations smooths both precipitation and temperature with large implications

in the representation of extremes. Moreover, large temporal differences in the number of stations within each grid box also adds another source of uncertainty since it can change trends of the time series (Hofstra et al., 2009; Frei, 2014; Beguería et al., 2016). Finally, in an analysis of the sources of uncertainty in observationally based gridded datasets, Herrera et al. (2018), highlight that the station density represents the major variability factor, irrespective of the interpolation method. The authors analysed several grids for Spain (complex topography) and Poland (smooth topography) and concluded that the influence of station density is more pronounced in Spain than in Poland due to the large

spatial variability and complex orography of the first.

The quality of the station observations is an additional source of observational uncertainty for gridded products. These uncertainties may be reduced by applying quality control procedures and homogenising the time series (Herrera et al., 2012). Precipitation time series also commonly suffer from undercatch associated to windy conditions, which usually results in underestimation of the correct precipitation rate (Frei et al., 2003). Yet in complex topography an increased uncertainty may be associated to the use of these types of corrections (Adam

et al., 2006). The areal representativeness of a particular station also poses a challenge. Again, in regions with high terrain gradients, like mountains or coastal areas, surface temperatures are affected by local circulations like sea-breeze, up/down slope breeze associated to nighttime radiative cooling in the valleys and to differentiated warming/cooling at sunrise/sunset of the slopes (Whiteman, 1982; Whiteman and McKee, 1982; Whiteman, 1990). Frei (2014) proposed a new interpolation method to tackle the latter, in which the thermal vertical profile of the station surrounding area is considered. Yet, Frei (2014) also acknowledges that the best way to reduce this type of uncertainty is through

high station density.

Recently, several national high-resolution grids have been compiled for individual European countries from dense observation networks: SAFRAN analysis at 8km grid spacing covering France at an hourly timestep (Durand et al., 1993; Quintana-Seguí et al., 2008; Vidal et al.,



2010), and its recently published extension for continental Spain and Balearic Islands (Quintana-Seguí et al., 2017); PTHBV, a 4km daily dataset for Sweden (Johansson, 2000; Johansson and Chen, 2003); the 5km resolution HYRAS for Germany (Rauthe et al., 2013; Frick et al., 2014); seNorge2 a daily dataset with an 1km resolution for Norway (Uboldi et al., 2008; Lussana et al., 2018); TabsD (MeteoSwiss, 2013a) and RhiresD (MeteoSwiss, 2013b) at 2km for Switzerland; CARPATCLIM a $0.1°$ grid covering parts of nine countries along the Carpathian

Mountains (Lakatos et al., 2013) and a $0.11°$ grid for Poland (Herrera et al., 2018).

In the Iberian Peninsula, Herrera (2011); Herrera et al. (2012) built a precipitation regular grid for continental Spain and Balearic Islands based on 2756 stations (Spain02) following the methodology used in E-OBS. The same methodology was also applied by Belo-Pereira et al. (2011) for continental Portugal using more than 400 stations (PT02). Both grids had a $0.2°$ resolution and a time span of 1950-2003. Recently, Herrera et al. (2015) updated the Spanish grid including precipitation and temperatures (daily maximum, mean and minimum)

and enhancing the spatial resolution to $0.1°$ (regular); moreover, they also provided results on a $0.11°$ rotated grid (CORDEX compliant) for the purpose of Regional Climate Model (RCM) evaluation. While the gridding methodology in the PT02 was the same as in Herrera et al. (2012), some discrepancies between the two datasets occurred near the borders, particularly in the northern mountains. These problems could be solved building a joint grid, using observational station data from both countries. Furthermore, the $0.2°$ resolution of the Portuguese grid is too coarse for regional climate studies and the lack of temperature grids also hinders a comprehensive analysis of the large climate

inter-annual and spatial variability characteristic of the Iberian climate (Esteban-Parra et al., 1998; Muñoz-Díaz and Rodrigo, 2004; Cardoso et al., 2013).

In this paper, we develop an Iberian wide daily regular grid at $0.1°$ resolution, for precipitation and temperatures (maximum, mean and minimum) as well as a $0.11°$ rotated grid (EURO-CORDEX compliant) suitable for model evaluation purposes. This grid is based on a high density network of stations across continental Portugal and Spain and Balearic Islands, with a reasonably stable number of stations for the

period 1971-2015. This represents the first precipitation and temperature gridded dataset for Iberia and can be considered an update of the PT02 dataset. Here, we also introduce the orography as covariate in the interpolation process, which was missing in the initial PT02 and Spain02. The resulting dataset is compared against the most recent version of E-OBS (v17.0, referred to as v17), which includes a new ensemble version (v17e) to assess observational uncertainty and allows for a new probabilistic intercomparisson of these datasets.

The paper is structured as follows: First, in Section 2 a description of the data and methods considered in this work is presented. Secondly,

the main results are described (Sec. 3). Finally, the main conclusions and discussions grown from the analysis are detailed in Section 4.

## 2   Data and Methods

### 2.1   Observation Network and Quality Control

The present work is based on a dense network of 3847 precipitation stations and over 380 temperature stations from the Spanish Agency of Meteorology (AEMET), the Portuguese Institute for Sea and Atmosphere (IPMA) and the Portuguese Environmental Agency (APA). The

final network was obtained applying the same quality control used to build *Spain02* (see Herrera, 2011; Herrera et al., 2012, for a detailed description of the dataset and the corresponding quality control), obtaining the observational network shown in Figure 1(a-b), including 3481 and 276 stations for precipitation and temperature, respectively. Figure 1(c) shows that there is a clear decline of the number of stations with available data in the last two decades, mainly for precipitation. Therefore, the resulting gridded product is not suitable for historical trend analysis, since biased results could be obtained as a result of the changing number of stations. Moreover, during the period 2009-2014 the

are very few precipitation stations in Portugal and, therefore, results should be interpreted with caution in this period. Overall, the spatial distribution of the stations is quite homogeneous over the Iberian Peninsula with a good representation of the orographical gradients, specially



for the case of precipitation (see the first column of Fig. 1). Therefore, the orography was included as a covariate in the interpolation process (at a monthly scale) to model and reflect these gradients.

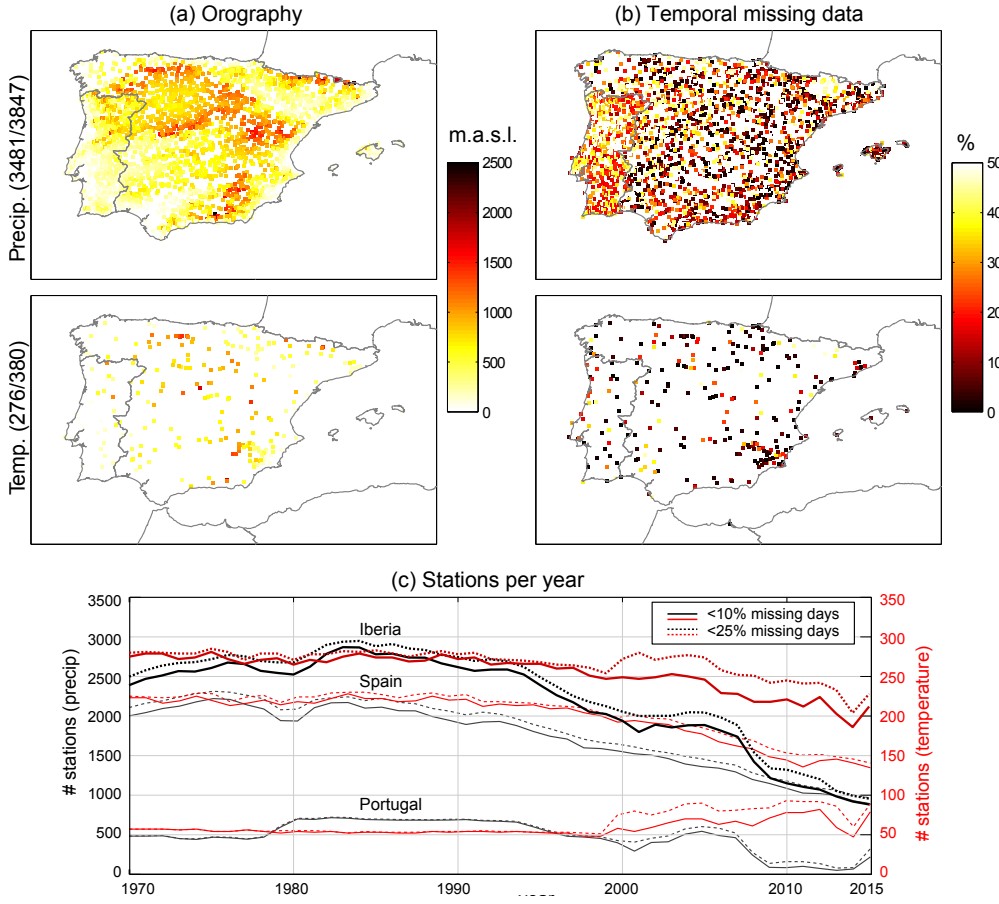

**Figure 1.** (a) Orography and (b) data availability (percentage of missing days) in the period 1971-2015 for precipitation (top) and temperature (bottom) for the observational networks considered for the interpolation. The numbers on the left of the figure reflect the initial and final number of stations ised, once the quality control is applied. (c) Number of stations – considering the Iberian Peninsula, Spain or Portugal – per year for different thresholds of annual missing data for precipitation (black) and temperature (red).

## 2.2 E-OBS Gridded Datasets (v17 and v17e)

E-OBS (Haylock et al., 2008) is the reference gridded dataset of daily precipitation and temperatures in Europe and has been previously used
5 to analyze the observational uncertainty in the context of the evaluation of regional climate models (see, e.g. Kotlarski et al., 2017). In this study, we use both the standard (v17, $0.2°$ resolution) and the ensemble (v17e, $0.1°$ resolution Cornes et al., 2018) versions of E-OBS v17 as benchmark for comparison purposes. In addition to the estimated daily value for each gridbox, the ensembles grid also provides a measure of daily uncertainty, characterized by the standard deviation of the ensemble. The first dataset is used for the sake of comparison with Iberia01





(see Fig. 2) and the second one (the ensemble) is used to assess the observational uncertainty provided by this dataset, by testing whether Iberia01 could be considered a realization of the E-OBS ensemble.

## 2.3  Weather Indices

In order to analyze the mean and extreme regimes of precipitation and temperature we use the indicators shown in Table 1. In particular,
the 50-year return value for each grid-box was used to characterize the extreme regimes (for the period 1971-2015) obtained by adjusting a Generalized Extreme Value (GEV) distribution to the series of annual maximum of daily values (see  Herrera et al., 2015, for a detailed description). In the case of precipitation, both wet-day frequency and rainfall intensity have been considered to properly characterize the mean regime.

**Table 1.** Precipitation and temperature indices used in this study.

| ID | Indicator | Units |
|----|-----------|-------|
| pr | Mean daily precipitation amount | mm/day |
| RR1 | Wet-day ($pr > 1mm$) frequency | % |
| RV50Yp | 50-years return value of daily precipitation | mm/day |
| tas | Mean daily 2-meters air temperature | deg. Celsius |
| RV50Yt | 50-years return value of the mean daily 2-meters air temperature | deg. Celsius |

## 2.4  Gridding Method

The Iberia01 daily gridded dataset for precipitation and temperatures was build using the previously described observational network and applying the area-averaged 3-dimensional (AA-3D) interpolation method described in previous studies (Herrera, 2011; Herrera et al., 2012, 2015). This interpolation method is an area-averaged method based on ordinary kriging (OK; Krige, 1951; Matheron, 1962) and 3-dimensional thin plate splines (3D-TPS; Craven and Wahba, 1979; Wahba, 1990; Hutchinson, 1998a, b) in a two-step process:

- first, the 3D-TPS is applied to the monthly value considering the orography as covariable;
- second, the daily anomaly is interpolated by applying OK;
- as a result, both the daily anomaly and monthly value are combined to obtain the interpolated daily values

In order to ensure the area-averaged representativity of the final values, the initial interpolation is done over an auxiliary $0.01°$ resolution grid and, then, the interpolated results are upscaled (averaged) to the target resolutions, in our case a regular version of $0.10°$ spatial resolution ($10km$ approx.) and a rotated version matching the grids considered in the EURO-CORDEX project ($0.11°$ and $0.44°$). In this work we only
describe for simplicity the regular version of $0.10°$ spatial resolution, although the other datasets are also provided (these datasets will be used in a future paper to evaluate the performance of EURO-CORDEX models over Iberia).



## 2.5 Effective Resolution

Taking into account the two-step interpolation procedure followed to develop the area-average representative gridded dataset, a natural doubt surges about the possible application of the auxiliary very-high resolution grid (1 km) to build a grid at a resolution higher than 10 km. For instance, the new RCM convective permitting simulations performed in the framework of the CORDEX Flagship Pilot Studies (FPS) reach a resolution of $2-3$ km and, thus, high resolution grids are needed for the evaluation of these projects (Giorgi et al., 2009; Jacob et al., 2014). In order to test this possibility, an illustrative example is considered: a convective high-resolution extreme precipitation event affecting the Iberian Peninsula occurred on 4-5 November 1997 characterized by heavy precipitation over most of the Iberian Peninsula, in particular crossing the Peninsula from the southwestern to the northeastern. This event had great socioeconomic impacts in Portugal (Ramos and Reis, 2002) and Spain (Lorente et al., 2008), and was ranked as the second greatest extreme precipitation event of the Iberian Peninsula (Ramos et al., 2015). We use this event as an illustrative case study in order to analyze the potential benefits provided by a 3 km gridded version of Iberia01, as compared with the standard 10 km resolution.

## 3 Results

Figure 2 shows the climatologies of the indices shown in Table 1 for Iberia02 ($0.1°$) and E-OBS (v17, $0.2°$ and v17e, $0.1°$). The three datasets provide similar results in the case of temperature, being the main differences located in the South, around the Guadalquivir and Guadiana basins, where the maximum values are attained. In addition, the uncertainty climatology (calculated as the temporal mean of the daily standard deviations of the ensemble values) is also provided for the ensemble version of E-OBS, showing a value around $2.5°$ in all the territory, with the exception of a number of kernels where the uncertainty is very small, corresponding to the stations used to build the E-OBS grid. Note that the uncertainty climatology is different in Spain and Portugal, with more clear kernels in the first case; this could be due to the different temporal coverage of both networks, with an increase of uncertainty due to days with no observation. Therefore, the uncertainty conveys relevant information on the station network used to build the grid.

In the case of precipitation, E-OBS is not able to reproduce neither the mean spatial pattern (pr) nor the intensity of the 50-year return value (RV50Yp). E-OBS underestimates mean precipitation by 15 - 20% (mean relative bias, for E-OBSv17 - v17e, respectively), particularly in the Central System range of the Iberian Peninsula, and 50-year return values by 42 - 47% (mean relative bias), with some very high biases in some Southern and Mediterranean regions. The case of wet-day frequency is different, since all datasets show a clear overestimation, with Iberia01 showing a more orographic pattern than E-OBS. In this case, the higher resolution of E-OBS v17e provides further spatial detail as compared to the standard v17 one, which is not evident for precipitation intensity. Moreover, the uncertainty (temporal mean of the daily standard deviations of the ensemble values) is of similar magnitude to the mean value (also with kernels of small uncertainty corresponding to stations) reflecting a large uncertainty for to this variable.

In order to quantitatively assess the differences between these two datasets, we use the "observational uncertainty" provided by the E-OBS v17e ensemble and test whether Iberia01 can be considered a realization of this ensemble (i.e. within the observational uncertainty range) with a certain confidence (90%) day by day. For this purpose, the E-OBS ensemble mean ($\mu$) and spread ($\sigma$) are used to define a normal distribution $N(\mu, \sigma)$ characterizing observational uncertainty for each grid box and day, and the corresponding Iberia01 values are classified as either inside or outside (values outside the P5-P95 percentile range) the uncertainty range for each grid box and day. Note that outsider values indicate significant differences between both datasets (as characterized by the E-OBS ensemble). Figure 3 shows the percentage of significantly different days for each gridbox, variable and season. For precipitation (first row), only Iberia01 wet-days were used in order to minimize the effect of the different wet-day frequencies. The differences for this variable exhibit a homogeneous spatial



**Figure 2.** Climatology of the different temperature (top) and precipitation (bottom) indices defined in Table 1, from the stations (local values), Iberia01 (0.1° resolution), E-OBS v17 (0.2°) and E-OBS v17e (0.1°), in rows. The mean daily standard deviation has been included for the latest dataset. The numbers show the spatial mean of each map.

pattern over the Peninsula with values around 10% in general; this is due to the large uncertainty of the daily E-OBS ensemble spread (see Fig. 2) thus questioning the practical utility of this measure of uncertainty for this variable. Regarding the temperatures, most of the spatial pattern presents values close to zero, reflecting the similarity between both datasets for these variables. However, some local differences are found particularly for the mean (second row) and maximum (third row) temperatures, with the greatest values reached in the Pyrenean and

Central ranges and the south coast of the Iberian Peninsula, in agreement with the differences shown in Figure 2. In this case, the ensemble uncertainty is in agreement with the differences between these two datasets found in Figure 2.

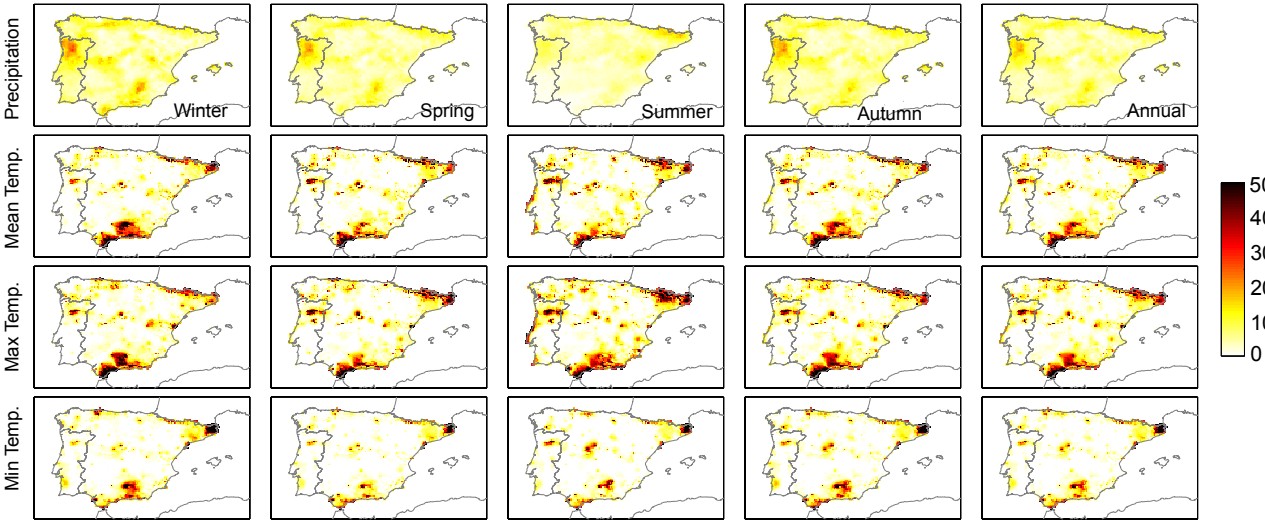

**Figure 3.** Percentage of significantly different days between Iberia01 and E-OBS v17e for each gridbox, variable and season, defined as the Iberia01 daily values outside the the $P5 - P95$ percentile interval of the normal distribution given by the E-OBS ensemble, in the period 1970-2015 for wet-days (first row), and mean (second row), maximum (third row) and (fourth row) minimum temperatures.

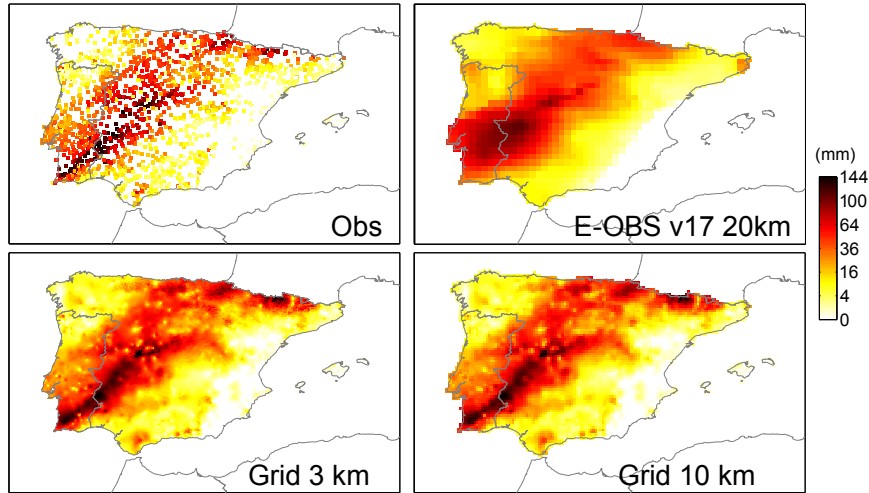

**Figure 4.** Daily precipitation of the 4-5 November 1997 observed and given by E-OBS v17, and a 3 km and 10 km version of Iberia02.





In order to explore the possibility to increase the resolution of the Iberia01 grid, we consider an extreme event occurred the 4-5 November 1997 and compare the resulting values of the $0.1°$ grid with a higher resolution $0.03°$ one developed using the auxiliary $0.01°$ grid generated in the interpolation process. Figure 4 shows the results obtained for the extreme event indicating that an increment of the Iberia01 resolution beyond 10 km has no clear impact in the effective resolution of the precipitation pattern. In particular, in spite of the clear improvement

of both versions of Iberia01 w.r.t. E-OBS v17 for all the parameters considered, there are only slight differences between both versions of Iberia01 when compared with observations.

Note that the interpolation method, independently on the target resolution, is calibrated to reproduce the spatial dependence of the mean field of the target variable, which is usually greater than the grid resolution ($1°$ approximately in this case). Therefore, the effective resolution of purely interpolated gridded products is limited by this spatial value, which define the size of the kernels used for the interpolation process.

As a result, in order to properly evaluate the convecting permitting CORDEX simulations, other approaches like regional reanalysis (e.g. Häggmark et al., 2000) or methods combining interpolation and analysis as the proposed by Quintana-Seguí et al. (2017) and Peral et al. (2017), among others, should be used.

## 4 Conclusions and Discussion

In this work a new gridded dataset for the Iberian Peninsula and the Balearic Islands based on a quality-controlled and dense station network

has been described and compared with E-OBS v17, considering both the standard and the ensemble version of this product, to reflect and analyze the observational uncertainty related with both datasets.

On the one hand, Iberia02 is able to reproduce the spatial pattern and intensity of both the mean and extreme regimes of precipitation and temperature, in terms of the weather indices defined in Table 1, including extreme events as the one occurred the 4-5 November 1997 shown in the Figure 4. For the weather indices considered, E-OBS v17 tends to underestimate the extremes and soften the spatial pattern of

precipitation, in agreement with other previous studies (Herrera et al., 2012). It is however more similar to Iberia02 in the case of temperature indices, with the main differences appearing in the Guadalquivir and Guadiana basins, and the Pyrenean range. In addition, although both datasets seem to reproduce more or less the same dry-days (see Figure 2) large differences appear when wet-days are considered, with E-OBS v17 identifying less than the 70% of the observed wet-days all around the Peninsula and falling up to the 40% in Summer.

On the other hand, considering the ensemble version of E-OBS, E-OBS v17e, an experimental framework to evaluate the observational

uncertainty has been defined, analizing if Iberia02 falls inside the ensemble given by E-OBS v17 and, then, if it could be considered a realization of the ensemble. In this case, we conclude that both datasets could be used indistinctly. First, note that the spread of the ensemble for precipitation has the same order of the mean value reflecting a large uncertainty for this variable in contrast to the one obtained for temperatures. In the case of precipitation the percentage of outliers for the wet-days ranges between 5% and 25% along the Peninsula whilst for temperatures most of the area show values less than 5%, with the regions identified previously (Guadalquivir basin, Pyrenees, etc.)

presenting the greatest percentages with values larger than 50% − 60%. These results are in agreement with the quantile distribution, with the temperature centered around the median, with some underestimation/overestimation in the case of the minimum/maximun temperatures, and the precipitation showing a clear overestimation of the quantiles that increases dramatically when only wet-days are considered. In summary, although in most of the domain the temperatures given by Iberia02 are included within the ensemble defined by E-OBS v17e, there are several regions with significant differences that should be considered/treated with caution. Moreover, in the case of precipitation both

datasets present significant differences that should be taken into account.



The Iberia01 dataset (Gutiérrez et al., 2019, DOI: http://dx.doi.org/10.20350/digitalCSIC/8641) is publicly available through the climate services portals of:

– IPMA: http://www.ipma.pt/pt/oclima/servicos.clima/

– AEMET: http://www.aemet.es/es/serviciosclimaticos/cambio_climat/datos_diarios





## 5 Code and data availability

All the datasets used in this work are publicly available. The Iberia01 dataset (Gutiérrez et al., 2019, DOI: http://dx.doi.org/10.20350/digitalCSIC/8641) is publicly available through the climate services portals of:

- IPMA: http://www.ipma.pt/pt/oclima/servicos.clima/

– DIGITAL.CSIC Open Science: http://hdl.handle.net/10261/183071

- Santander User Data Gateway (UDG): http://meteo.unican.es/tds5/dodsC/Iberia01/Iberia01_v1.0_010reg_aa_3d.ncml

The E-OBS v17 dataset is remotely available through the KNMI's THREDDS Data Server http://opendap.knmi.nl/knmi/thredds/e-obs/e-obs-catalog.html and the ensemble version E-OBS v17e is available through the Copernicus' Climate Change Service http://surfobs.climate.copernicus.eu/dataaccess/access_eobs.php.

The R code to reproduce the comparison analysis between the Iberia01 and E-OBS v17 dataset is publicly available from the GitHub repository of Santander Meteotology Group: https://github.com/SantanderMetGroup/notebooks

*Author contributions.* Herrera S., Gutiérrez J.M. and Soares P.M. conceived the study; Gutiérrez J.M., Soares P.M., Cardoso R.M., Espíritu-Santo F. and Viterbo P, obtained and processed the Spanish and Portuguese observational datasets; Herrera S. implemented the code to make the interpolation and the analysis, and built the dataset and figures of the paper; Herrera S., Soares P.M., Gutiérrez J.M. and Cardoso R.M.

wrote the manuscript and all the authors revised the results.

*Competing interests.* The authors declare that there are not any competing interest.

*Acknowledgement.* This work was partially funded by the Spanish Government R&D Programme (Exp. CGL2010-21869 and CGL2010-22158-C02). Pedro M.M. Soares and Rita M. Cardoso wish to acknowledge the SOLAR (PTDC/GEOMET/7078/2014) project and the funding by the project FCT UID/GEO/50019/2019 - Instituto Dom Luiz.

The authors are grateful to the Portuguese Institute for Sea and Atmosphere (IPMA) and the Portuguese Environmental Agency for providing the needed observational data. The authors are also grateful to AEMET for providing the necessary data to do this work. We acknowledge the E-OBS dataset from the EU-FP6 project ENSEMBLES (http://ensembles-eu.metoffice.com) and the data providers in the ECA&D project (http://www.ecad.eu). We acknowledge the E-OBS dataset from the EU-FP6 project UERRA (http://www.uerra.eu) and the Copernicus Climate Change Service, and the data providers in the ECA&D project (https://www.ecad.eu)





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
