# Peer review of "Iberia01: A new gridded dataset of daily precipitation and temperatures over Iberia"

_Earth System Science Data, 2019_

## Referee Comment (RC1) · Anonymous Referee #1 · 19 Jul 2019

Review of "Iberia01: A new gridded dataset of daily precipitation and temperatures over Iberia"

This paper presents a gridded climatological data product for the Iberian peninsula called "Iberia01" which appears to be a revision of a previous data product called "Spain02", using the same station network and interpolation methods, except including "orography" (elevation?) as covariate in the thin-plate spline step. Iberia01 is compared against a standard (E-OBS), finding more or less similar predictions except in some specific locations for specific climate variables. For precipitation it is found that the higher-resolution Iberia01 predictions show more small-scale variation than the coarse E-OBS product, at least in the case of a specific major precipitation event. The construction and integrity of the dataset appear to be well-done overall, although much

of the methods refer to a previously published paper by the same authors. The figures are well-done as well. As a presentation of a new dataset, I think this paper should suffice (with some major revisions, clarifications, etc), although it is not clear whether any of the techniques or analyses are particularly novel.

specific lines referenced as (pg:line)

Major Issues:

None of the links (p11:4-6) to the data worked for me. The first sent me to a generic landing page in Portuguese. The second sent me to a site where the data was embargoed and required a login account. The third sent an error message. I assume based on the R code that the third access point requires authentication with Santander and requests via a specialized R function. The AEMET link (pg 10) sent me to another landing page where it was unclear how to find the dataset. Either way, I could not access the data.

The authors state that the E-OBS dataset is taken as a benchmark (4:7) but later claim that the dataset is biased for key variables (6:22). It is not clear in the methods how this assessment is made or quantified.

It would be nice to provide some ideas explaining the specific deviations (e.g. along the coast) between Iberia01 and E-OBS in the discussion

The assessment of resolution for the convective rain event is unsatisfying. First, why do the authors present the 20 km product (v17) and not the 10 km product (v17e) for E-OBS, which seems like a much better comparison? Second, the authors claim that the difference in resolution for Iberia01 10 vs 3 km resolution does not matter, but this is not quantitatively examined or explained in any way. Are the authors using gestalt, I assume?

There are numerous grammatical errors, run-on sentences and awkward phrasings throughout. I have noted some below, but not all. The writing is good in terms of
logic, but needs a careful proofread (possibly by a native English speaker) before it is publishable.

The methods frequently refer to Herrera 2011, 2012. However, more brief descriptions of these methods would be helpful, such as the QC protocols.

Minor issues:

Where did the 'orography' dataset come from? Isn't this just elevation?

The R code (pg 11, line 10) Seems to only calculate and visualize climatologies but doesn't actually do any direct comparisons.

(6:10) How exactly?

paragraph (6:29ff) move to methods

(6:30): "can be considered a realization of" seems like an awkward way to phrase it. Why not 'differs significantly from'?

(7:2) "thus questioning" – Move interpretations like this to the discussion and flesh them out. I would tend to disagree with this statement as stands.

Technical issues:

Please remove "In order" from all sentences beginning with "In order to", as this is redundant.

(1:9) Run on sentence

(1:11) omit "As a result"

(1:15) rephrase

(2:6) expands -> includes

(2:11-12) reference for this assertion?

[Figure]

(2:14) omit 'the'

(2:19) 'smooths' awkward term here

(3:6) include 'and' between citations

(3:15) runon sentences

(3:20) Is this really the first? Seems like there are others, referenced in the same sentence (PT02)

(4:8) Replace 'first dataset', 'this one' etc with specific title of each. Confusing

(5:10) "temperature was built"

(6:6) run-on

(6:13) How were clims aggregated?

(6:14) "the main differences being"

(6:29) "we used"

(9:7) on -> of

(9:28) and (9:30) – these sentences are both difficult to understand.

---

## Referee Comment (RC2) · Shawn Marshall (Referee) · 7 Aug 2019

The authors present an extensive, long-term dataset of temperature and precipitation in Iberia, based on a combination and extension of datasets from Spain and Portugal. While it is not dramatically new from previous data compilations by the senior author and his colleagues, they do introduce higher resolution and some new analysis. For instance, having elevation as a covariate in the interpolation procedure is a valuable improvement.

While incremental, this is a valuable dataset that can be used in a wide range of applications. I don't know of a network of observations this extensive, dense, and long-running anywhere in the world. While it is a shame to see the number of observations

degrade in recent years, this is a valuable dataset that can be used for either weather or climate analyses. I can certainly see the value of this dataset as a test of the Cordex high-resolution simulations. The paper is well-written: clear and concise. I recommend publication with minor revisions.

Minor errors or clarifications

p.2,l.5, "higher longitudinal and latitudinal resolution", I think?

p.2,l.14, should be "has been analyzed"

Figure 1 caption, "ised" should be "used"

Table 1, RV50Yt - shouldn't this be the maximum daily 2-m air temperature?

p.6,l.8,"southwest to the northeast"

p.6,l.14, "with" the main differences being...

Discussion of Figure 2. It is hard to discern the differences. Difference maps would help to illustrate the main differences of interest between the datasets.

p.6,l.24, "all datasets show a clear overestimation" - why do you think this is? I don't understand why this would be for wet-day frequency, as it seems that this should come in a straightforward way from the dataset. How does interpolation or modelling introduce too many wet-days?

It would be interesting to see mean precipitation here as well, for each dataset.

p.9, conclusions - the authors frequently refer to Iberia02, but that is the next paper isn't it? Up to here and in the title this is presenting Iberia01.

p.9, ll.22-23 - I must misunderstand wet-days. I don't understand how the dry-days could be equivalent between datasets but the wet-days differ; I would have thought that wd = 365 - dd. This is likely just my deficiency, but others might also be confused here so some explanation would be good.

p.6,l.21, double negative - I think it should be "either" and "or"

---

## Author Comment (AC1) · 16 Aug 2019

**Anonymous Referee #1:**

*Review of "Iberia01: A new gridded dataset of daily precipitation and temperatures over Iberia"*
*This paper presents a gridded climatological data product for the Iberian peninsula called "Iberia01" which appears to be a revision of a previous data product called "Spain02", using the same station network and interpolation methods, except including "orography" (elevation?) as covariate in the thin-plate spline step. Iberia01 is compared against a standard (E-OBS), finding more or less similar predictions except in some specific locations for specific climate variables. For precipitation it is found that the higher-resolution Iberia01 predictions show more small-scale variation than the coarse E-OBS product, at least in the case of a specific major precipitation event. The construction and integrity of the dataset appear to be well-done overall, although much of the methods refer to a previously published paper by the same authors. The figures are well-done as well. As a presentation of a new dataset, I think this paper should suffice (with some major revisions, clarifications, etc), although it is not clear whether any of the techniques or analyses are particularly novel.*

*Interactive comment on Earth Syst. Sci. **Data Discuss.**, https://doi.org/10.5194/essd-2019-95, 2019.*

*specific lines referenced as (pg:line)*

**Response:** We thank the reviewer for the comments and the time devoted to our paper. Please, see below our point-by-point responses and the changes highlighted as tracked changes in the new version of the manuscript.

**Major Issues:**

*None of the links (p11: 4-6) to the data worked for me. The first sent me to a generic landing page in Portuguese. The second sent me to a site where the data was embargoed and required a login account. The third sent an error message. I assume based on the R code that the third access point requires authentication with Santander and requests via a specialized R function. The AEMET link (pg 10) sent me to another landing page where it was unclear how to find the dataset. Either way, I could not access the data.*

**Response:** We thank the referee for point out this comment. First, the embargo was established in order to prevent the use of the dataset before the publication of the corresponding reference. We have requested to avoid the embargo of the dataset in order to make it publicly available. Second, although this will be the main access point, we are trying to give access to the dataset also through the National services referred in the paper, IPMA and/or AEMET.

*The authors state that the E-OBS dataset is taken as a benchmark (4:7) but later claim that the dataset is biased for key variables (6:22). It is not clear in the methods how this assessment is made or quantified.*

**Response:** As the referee has pointed out, E-OBS was defined as benchmark in the current version of the manuscript cause this dataset is considered the reference at European scale. However, as has been reflected in several studies (Belo-Pereira et al. 2011; Turco and Llasat 2011; Flaounas et al. 2012; Herrera et al. 2012 and 2016; Turco et al. 2013; Prein and Gobiet 2016), at national or regional scale E-OBS presents some known biases, mainly in regions with complex orography and/or with low stations' density. To quantify the differences of the spatial pattern for each parameters the following table has been included in the manuscript reflecting several verification parameters considering the nearest grid box of each dataset to the local station observations. In addition, the text has been rewritten accordingly.

| Iberia01 | tas | RV50Yt | pr | RV50Yp | RR1 |
|---|---|---|---|---|---|
| MAE | 0.5404 | 1.6162 | 0.2888 | 20.1838 | 11.2783 |
| BIAS | -0.2099 | -1.3145 | 0.0881 | -17.8175 | 11.2763 |
| RMSE | 0.8902 | 2.9837 | 0.5651 | 28.1075 | 13.6863 |
| Correlation | 0.9422 | 0.7319 | 0.8496 | 0.8623 | 0.4304 |
| **E-OBS v17** | **tas** | **RV50Yt** | **pr** | **RV50Yp** | **RR1** |
| MAE | 0.8212 | 2.6219 | 0.4288 | 42.4205 | 10.0718 |
| BIAS | -0.2603 | -2.1310 | -0.2703 | -42.2184 | 9.9931 |
| RMSE | 1.1530 | 3.9377 | 0.7510 | 54.2519 | 12.5221 |
| Correlation | 0.8931 | 0.5403 | 0.7508 | 0.5891 | 0.4297 |
| **E-OBS v17e** | **tas** | **RV50Yt** | **pr** | **RV50Yp** | **RR1** |
| MAE | 0.8260 | 2.6720 | 0.4357 | 46.9555 | 11.4778 |
| BIAS | -0.3341 | -2.3033 | -0.3021 | -46.7663 | 11.4641 |
| RMSE | 1.1811 | 4.0530 | 0.7600 | 58.0514 | 13.7543 |
| Correlation | 0.9047 | 0.5365 | 0.7560 | 0.5659 | 0.4374 |

*Table 1: Comparison between the spatial pattern of the different gridded datasets against the observations for the indices considered.*

Belo-Pereira, M., Dutra, E., and Viterbo, P.: Evaluation of global precipitation data sets over the Iberian Peninsula, Journal of Geophysical Research: Atmospheres, 116, 1–16, doi:10.1029/2010JD015481, 2011.

Flaounas, E., Drobinski, P., Borga, M., Calvet, J.-C., Delrieu, G., Morin, E., Tartari, G., and Toffolon, R.: Assessment of gridded observations
used for climate model validation in the Mediterranean region: the HyMeX and MED-CORDEX framework, Environmental Research
Letters, 7, 024 017, doi:10.1088/1748-9326/7/2/024017, 2012

Herrera, S., Gutiérrez, J. M., Ancell, R., Pons, M. R., Frías, M. D., and Fernández, J.: Development and analysis of a 50-year high-resolution
daily gridded precipitation dataset over Spain (Spain02), International Journal of Climatology, 32, 74–85, doi:10.1002/joc.2256, 2012.

Herrera, S., Fernández, J., and Gutiérrez, J. M.: Update of the Spain02 gridded observational dataset for EURO-CORDEX evaluation: assessing the effect of the interpolation methodology, International Journal of Climatology, 36, 900–908, doi:10.1002/joc.4391, 2016.

Prein, A. F. and Gobiet, A.: Impacts of uncertainties in European gridded precipitation observations on regional climate analysis, International Journal of Climatology, pp. n/a–n/a, doi:10.1002/joc.4706, 2016.

Turco, M., Zollo, a. L., Ronchi, C., De Luigi, C., and Mercogliano, P.: Assessing gridded observations for daily precipitation extremes in the Alps with a focus on northwest Italy, Natural Hazards and Earth System Science, 13, 1457–1468, doi:10.5194/nhess-13-1457-2013, 2013.

Turco, M. and Llasat, M. C.: Trends in indices of daily precipitation extremes in Catalonia (NE Spain), 1951-2003, Natural Hazards and Earth System Science, 11, 3213–3226, doi:10.5194/nhess-11-3213-2011, 2011.

*It would be nice to provide some ideas explaining the specific deviations (e.g. along the coast) between Iberia01 and E-OBS in the discussion.*

**Response:** *We have included a paragraph discussing this point a pointing out to possible reason of the observed differences between both datasets:*

*"Note that the complex orography and the influence of both the Atlantic Ocean and the Mediterranean Sea modulate the precipitation over the Iberian Peninsula, leading to particular regimes, as the cold drop in the east coast, that a continental adjustment of the interpolation model is not able to reproduce, even more when a low-dense observational network is considered. In this sense, the large increase of rain gauges considered in Iberia01, when compared with E-OBS, give rise to a much improved precipitation rendering. In the case of temperature, although the observational network considered is similar in both cases, the pattern tends to be more orographic in E-OBS v17 due to the continental adjusment of the interpolation method that overrates this component avoiding regional behaviors. In addition, the contribution of the obersvational network considered in France also has a clear effect on the interpolated value over the Pyrennes and the northeast of the Iberian Peninsula."*

*The assessment of resolution for the convective rain event is unsatisfying. First, why do the authors present the 20 km product (v17) and not the 10 km product (v17e) for E-OBS, which seems like a much better comparison?*

*Second, the authors claim that the difference in resolution for Iberia01 10 vs 3 km resolution does not matter, but this is not quantitatively examined or explained in any way. Are the authors using gestalt, I assume?*

**Response:** *Following the referee's comment we have updated the figure including the high-resolution version of E-OBS (~10 km) but, as can be seen in the following figure, the conclusions have not changed.*
  **Figure:** *Comparison between both resolutions of the E-OBS v17e dataset.*

[Figure]

In addition, we have obtained several parameters to compare the different datasets and resolutions. Table 2 shows the results of the comparison against the observations reflecting that the new dataset performs better than E-OBS v17 for all the parameters considered.

To evaluate the effect of the resolution we have compared both resolutions of each dataset considering the same parameters.

| Measure | Iberia ~3 km | Iberia ~10 km | E-OBS v17 ~25 km | E-OBS v17e ~10 km |
|---|---|---|---|---|
| MAE | 3.3861 | 4.8142 | 11.0998 | 10.9936 |
| BIAS | 0.2352 | 0.7046 | -1.7382 | -1.4243 |
| RMSE | 6.5430 | 8.9360 | 19.0300 | 18.7614 |
| Correlation | 0.9746 | 0.9522 | 0.7625 | 0.7691 |

**Table 2:** Comparison between the spatial pattern of the different gridded datasets against the observations.

To evaluate the effect of the resolution we have compared both resolutions of each dataset considering the spatial pattern and his statistical distribution. In both cases, the Pearson correlation is greater than 0.98 and statistically significant at 95% of significance. Moreover, the hypothesis test applied to compare the mean and the variance of the spatial patterns reflects that both resolutions come from a distribution with the same mean and variance.

| Measure | Iberia ~3 km | Iberia ~10 km | E-OBS v17 ~25 km | E-OBS v17e ~10 km |
|---|---|---|---|---|
| Correlation | 0.9855 | | 0.9912 | |
| H (mean) | 0 | | 0 | |
| H (std) | 0 | | 0 | |

**Table 3:** Comparison between the spatial pattern of the different resolutions considered to analyse the effective resolution of the gridded datasets.

When both E-OBS v17 and Iberia gridded dataset are compared in terms of spatial correlation, the Pearson coefficient falls to 0.77-0.78, depending on the resolutions considered, reflecting that E-OBS v17 is not able to reproduce the spatial pattern and that the effective resolution of both datasets is 0.1º and 0.25º for Iberia and E-OBS v17, respectively.

There are numerous grammatical errors, run-on sentences and awkward phrasings throughout. I have noted some below, but not all. The writing is good in terms of logic, but needs a careful proofread (possibly by a native English speaker) before it is publishable.

*Response: We have revised the text following the referee's comment.*

*The methods frequently refer to Herrera 2011, 2012. However, more brief descriptions of these methods would be helpful, such as the QC protocols.*

*Response: We have extended the description of the quality control procedure in the new version of the manuscript. In addition, two files, one for precipitation and other for temperature, reflecting the observational networks used to build the gridded dataset and their main properties in terms of missing data have been included in the server distributing the dataset.*

**Minor Issues:**

*Where did the 'orography' dataset come from? Isn't this just elevation?*

*Response: We have considered the orography given by the Global Digital Elevation Model (GTOPO30) which provides gridded 30 arc seconds (~1 km) elevation for the world (http://webhelp.esri.com/arcgisdesktop/9.3/index.cfm?TopicName=Global_Digital_Elevation_Model_(GTOPO30)).*

*The R code (pg 11, line 10) Seems to only calculate and visualize climatologies but doesn't actually do any direct comparisons.*

*Response: We agree with the referee and we have clarify this point in the text.*

*(6:10) How exactly?*

*Response: We have rewritten the sentence to clarify this point and included the corresponding Table with the results:*

*"We use this event as an illustrative case study in order to analyze the potential benefits provided by a 3 km gridded version of Iberia01, as compared with the standard 10 km resolution, in terms of the spatial (Pearson) correlation, and the comparison of the mean and variance of the spatial patterns through the Student's t-test and Snedecor's F-test, respectively, for two samples, corresponding to the low- and high-resolution versions of both E-OBS and Iberia01 gridded datasets."*

*paragraph (6:29ff) move to methods*

*Response: We have modified the manuscript accordingly and included this paragraph at the end of the section "E-OBS Gridded Datasets (v17 and v17e)".*

*(6:30): "can be considered a realization of" seems like an awkward way to phrase it. Why not 'differs significantly from'?*

*Response: We have modified the sentence accordingly.*

*(7:2) "thus questioning" – Move interpretations like this to the discussion and flesh them out. I would tend to disagree with this statement as stands.*

*Response: We have modified the sentence accordingly.*

**Technical issues:**

*Please remove "*In order*" from all sentences beginning with "*In order to*", as this is redundant.*

**Response:** *We have modified the manuscript accordingly.*

*(1:9) Run on sentence*

**Response:** *We have rewritten the sentence accordingly:*

*"Finally, the possibility to increase the resolution of the dataset using the same interpolation approach is analyzed considering an extreme event of convective precipitation affecting the Iberian Peninsula and the auxiliary very high resolution grid (0.01º), built during the interpolation process to obtain the area-average representativity of the final dataset."*

*(1:11) omit "*As a result*"*

**Response:** *We have modified the sentence accordingly.*

*(1:15) rephrase*

**Response:** *The paragraph has been completely reformulated:*

*"We show that Iberia01 produces more realistic patterns than E-OBS v17 in the case of precipitation for all the indices considered, although both are comparable for temperatures. These differences were assessed using a probabilistic approach based on the E-OBS ensemble. For precipitation, significant differences ---at a 10% level--- between both datasets were found for less than 25% of days over the Iberian Peninsula. For temperature, a very inhomogeneous pattern was obtained, with either a small (in most of the regions) or large fraction of significantly different days. The great uncertainty of the precipitation given by E-OBS ensemble, in which the standard deviation of the ensemble has the same order than the mean value, increases the significance of the results obtained for this variable reflecting the differences between both datasets."*

*(2:6) expands -> includes*

**Response:** *We have modified the sentence accordingly.*

*(2:11-12) reference for this assertion?*

**Response:** *We have included the citation's number of E-OBS in Scopus (1491 citations, in contrast with the 182 of Spain02 (Herrera et al. 2012) and the 94 of PT02 (Belo-Pereira et al. 2011)) in order to justify the assertion. In addition, we have modified the sentence to better clarify its meaning:*
*"With more than 1490 citations in Scopus (at August 2019), this is the most used climate reference for European climate studies (e.g. in the Iberian Peninsula, Spain02 and PT02 have more than 180 and 90 citations, respectively)"*

*(2:14) omit 'the'*

**Response:** *We have modified the sentence accordingly.*

*(2:19) 'smooths' awkward term here*

**Response:** We have rewritten the sentence accordingly: *"… , a reduction in the density of stations decreases the variability of both precipitation and temperature with large implications in the representation of extremes."*

*(3:6) include 'and' between citations*

**Response:** *We have modified the sentence accordingly.*

*(3:15) Run on sentences*

**Response:** *We have modified the sentence:*

*"Furthermore, the 0.2º resolution of the Portuguese grid is too coarse for regional climate studies. On the other hand, the lack of temperature grids also hinders a comprehensive analysis of the large inter-annual and spatial variability, characteristic of the Iberian climate. These problems could be solved building a joint grid, using observational station data from both countries."*

*(3:20) Is this really the first? Seems like there are others, referenced in the same sentence (PT02)*

**Response:** *The Iberia01 dataset is the first gridded dataset built over the Iberian Peninsula, not focused on a greater domain as E-OBS, and reaching the proposed spatio-temporal resolution. Other products have a greater spatial resolution but contain only climatologies or are limited to particular regions of the Iberian Peninsula. We have rewritten the sentence to clarify this point:*

*"This represents the first gridded dataset of daily precipitation and temperatures focused on Iberia, and can be considered an update of the PT02 dataset."*

*(4:8) Replace 'first dataset', 'this one' etc with specific title of each. Confusing*

**Response:** *We have modified the sentence accordingly.*

*(5:10) "temperature was built"*

**Response:** *We have modified the sentence accordingly.*

*(6:6) Run on sentence*

**Response:** *We have modified the sentence accordingly:*

*"To test this possibility, an illustrative example is considered: a convective high-resolution extreme precipitation event occurred on 4-5 November 1997, and characterized by heavy precipitation over most of the Iberian Peninsula."*

*(6:13) How were clims aggregated?*

**Response:** The climatologies have been obtained averaging the annual values of the indices (tas, pr and RR1). In the case of the 50-years return value the index is representative of all the period, so it is its own climatological value.

*(6:14) "the main differences being"*

*Response: We have modified the sentence accordingly.*

*(6:29) "we used"*

**Response:** We have modified the sentence accordingly.

*(9:7) on -> of*

**Response:** We have modified the sentence accordingly.

*(9:28) and (9:30) – these sentences are both difficult to understand.*

**Response:** We have modified the sentence accordingly.

"In the case of precipitation (Figure 3, first row), the percentage of outliers considering only the wet-days ranges between 5% and 25% along the Peninsula. For temperatures (Figure 3, second to fourth rows), most of the area shows percentages less than 5% of outliers, with only some regions previously identified (Guadalquivir basin, Pyrenees, etc.) presenting values larger than 50%-60%. In summary, although in most of the domain the temperatures given by Iberia01 are included within the ensemble defined by E-OBS v17e, there are several regions with significant differences that should be considered/treated with caution. Moreover, in the case of precipitation both datasets present significant differences that should be taken into account."

The sentence "These results are in agreement with the quantile distribution, with the temperature centered around the median, with some underestimation/overestimation in the case of the minimum/maximun temperatures, and the precipitation showing a clear overestimation of the quantiles that increases dramatically when only wet-days are considered." made reference to a figure that was removed from the final version of the paper, so we have removed also the sentence in the new version of the manuscript.

**Iberia01: A new gridded dataset of daily precipitation and temperatures over Iberia**

Sixto Herrera[a], Rita M. Cardoso[b], Pedro M.M. Soares[b], Fátima Espírito–Santo[c], Pedro Viterbo[c], and José M. Gutiérrez[d]

[a]Meteorology Group. Dept. of Applied Mathematics and Computer Science. Universidad de Cantabria. Santander, Spain
[b]Instituto Dom Luiz (IDL), Facultade de Ciências, Universidade de Lisboa, Lisboa, Portugal
[c]Instituto Português do Mar e da Atmosfera (IPMA), Lisboa, Portugal
[d]Meteorology Group. Instituto de Física de Cantabria, CSIC-University of Cantabria, Santander, Spain

**Correspondence:** Sixto Herrera (sixto.herrera@unican.es)

**Abstract.** The present work introduces a new observational gridded dataset produced using a dense network (thousands) of stations over the Iberian Peninsula (referred to as Iberia01, Gutiérrez et al. (2019), DOI: http://dx.doi.org/10.20350/digitalCSIC/ 8641), providing daily precipitation and temperatures for the period 1971-2015 at $0.1°$ regular (and $0.11°$ rotated CORDEX compliant) resolutions. A comparison with both the standard and ensemble version of the E-OBS v17 dataset (at $0.25°$ and $0.1°$ resolutions, respectively) is undertaken in order to assess observational uncertainty in this region. First, a standard comparison is performed for several weather indices, obtaining the differences between both datasets. Secondly, a new probabilistic intercomparison analysis is introduced, using the E-OBS ensemble (v17e) to characterize observational uncertainty and testing the hypothesis that Iberia01 is a realization of the ensemble (i.e. it falls within the observational uncertainty range provided by E-OBS). Finally, the possibility to increase the resolution of the dataset using the same interpolation approach is analyzed considering an extreme event of convective precipitation affecting the Iberian Peninsula and the auxiliary very high resolution grid ($0.01°$), built during the interpolation process to obtain the area-average representativity of the final dataset.

We show that Iberia01 produces more realistic patterns than E-OBS v17 in the case of precipitation for all the indices considered, although both are comparable for temperatures. These differences were assessed using a probabilistic approach based on the E-OBS ensemble. For precipitation, significant differences —at a 10% level— between both datasets were found for less than 25% of days over the Iberian Peninsula. For temperature, a very inhomogeneous pattern was obtained, with either a small (in most of the regions) or large fraction of significantly different days. The great uncertainty of the precipitation given by E-OBS ensemble, in which the standard deviation of the ensemble has the same order than the mean value, increases the significance of the results obtained for this variable reflecting the differences between both datasets.

KEY WORDS: *Observational uncertainty; E-OBS; ensemble; gridded observations; kriging; thin plate splines; extremes; Clime; Precipitation; Temperature*

*Copyright statement.* The Iberia01 gridded dataset is made available under the Open Database License. Any rights in individual contents of the database are licensed under the Database Contents License.

[revised manuscript text omitted]

standard deviations of the ensemble values) is of similar magnitude to the mean value (also with kernels of small uncertainty corresponding to stations) reflecting a large uncertainty for this variable.

Figure 3 shows the percentage of significantly different days for each gridbox, variable and season. For precipitation (first row), only Iberia01 wet-days were used in order to minimize the effect of the different wet-day frequencies. The differences for this variable exhibit a homogeneous spatial pattern over the Peninsula with values around 10% in general; this is due to the large uncertainty of the daily E-OBS ensemble spread (see Fig. 2). Regarding the temperatures, most of the spatial pattern presents values close to zero, reflecting the similarity between both datasets for these variables. However, some local differences are found particularly for the mean (second row) and maximum (third row) temperatures, with the greatest values reached in the Pyrenean and Central ranges and the south coast of the Iberian Peninsula, in agreement with the differences shown in Figure 2. In this case, the ensemble uncertainty is in agreement with the differences between these two datasets found in Figure 2.

To explore the possibility to increase the resolution of the Iberia01 grid, we consider the extreme event occurred the 4-5 November 1997 and compare the resulting values of the $0.1°$ grid with a higher resolution $0.03°$ one developed using the auxiliary $0.01°$ grid generated in the interpolation process. Tables 3 and 4, Figure 4 show the results obtained for the extreme event indicating that an increment of the Iberia01 resolution beyond 10 km has no clear impact in the effective resolution of the precipitation pattern. In particular, in spite of the clear improvement of both versions of Iberia01 w.r.t. E-OBS v17e for all the parameters considered (see Table 3), there are only slight differences between both versions of Iberia01 when compared with observations.

Moreover, as it is reflected in Table 4, the spatial correlation between both resolutions is greater than 0.98 for both datasets, and any significant — at 5 % level — difference is found for the mean and variance of the spatial pattern according to the applied hypothesis test for two independent samples, the Student's t-test and Snedecor's F test for the mean and the variance, respectively.

[Figure]

**Figure 3.** Percentage of significantly different days between Iberia01 and E-OBS v17e for each gridbox, variable and season, defined as the Iberia01 daily values outside the the $P5 - P95$ percentile interval of the normal distribution given by the E-OBS ensemble, in the period 1970-2015 for wet-days (first row), and mean (second row), maximum (third row) and (fourth row) minimum temperatures.

[Figure]

**Figure 4.** Daily precipitation of the 4-5 November 1997 observed and given by E-OBS v17e, and a 3 km and 10 km version of Iberia01.

Note that the interpolation method, independently of the target resolution, is calibrated to reproduce the spatial dependence of the mean field of the target variable, which is usually greater than the grid resolution ($1°$ approximately in this case). Therefore, the effective resolution of purely interpolated gridded products is limited by this spatial value, which define the size of the kernels used for the interpolation process. As a result, in order to properly evaluate the convecting permitting CORDEX simulations, other approaches like regional reanalysis (e.g.

**Table 3.** Comparison between the spatial pattern of the different gridded datasets against the observations.

| Measure | Iberia01 3 km | Iberia01 10 km | E-OBS v17 (25 km) | E-OBS v17e (10 km) |
|---------|---------------|----------------|-------------------|--------------------|
| MAE | 3.3861 | 4.8142 | 11.0998 | 10.9936 |
| BIAS | 0.2352 | 0.7046 | -1.7382 | -1.4243 |
| RMSE | 6.5430 | 8.9360 | 19.0300 | 18.7614 |
| CORR | 0.9746 | 0.9522 | 0.7625 | 0.7691 |

Häggmark et al., 2000) or methods combining interpolation and analysis as the proposed by Quintana-Seguí et al. (2017) and Peral et al. (2017), among others, should be used.

**Table 4.** Comparison between the spatial pattern of the different resolutions considered to analyse the effective resolution of the gridded datasets.

| Measure | Iberia01 3 km vs. Iberia01 10 km | E-OBS v17 (25 km) vs. E-OBS v17e (10 km) |
|---------|----------------------------------|-------------------------------------------|
| CORR | 0.9855 | 0.9912 |
| t test (H) | 0 | 0 |
| F test (H) | 0 | 0 |

**4 Conclusions and Discussion**

In this work a new gridded dataset for the Iberian Peninsula and the Balearic Islands based on a quality-controlled and dense station network
5   has been described and compared with E-OBS v17, considering both the standard and the ensemble version of this product, to reflect and analyze the observational uncertainty related with both datasets.

On the one hand, Iberia01 is able to reproduce the spatial pattern and intensity of both the mean and extreme regimes of precipitation and temperature, in terms of the weather indices defined in Table 1, including extreme events as the one occurred the 4-5 November 1997 shown in the Figure 4. For the weather indices considered, E-OBS v17 tends to underestimate the extremes and soften the spatial pattern of
10   precipitation, in agreement with other previous studies (Herrera et al., 2012). It is however more similar to Iberia01 in the case of temperature indices, with the main differences appearing in the Guadalquivir and Guadiana basins, and the Pyrenean range. In addition, both datasets present large differences for wet-days (see Figure 2), with E-OBS v17 identifying less than the 70% of the observed wet-days all around the Peninsula and falling up to the 40% in Summer. Note that the complex orography and the influence of both the Atlantic Ocean and the Mediterranean Sea modulate the precipitation over the Iberian Peninsula, leading to particular regimes, as the cold drop in the east coast,
15   that a continental adjustment of the interpolation model is not able to reproduce, even more when a low-dense observational network is considered. In this sense, the large increase of rain gauges considered in Iberia01, when compared with E-OBS, give rise to a much improved precipitation rendering. In the case of temperature, although the observational network considered is similar in both cases, the pattern tends to be more orographic in E-OBS v17 due to the continental adjusment of the interpolation method that overrates this component avoiding

regional behaviors. In addition, the contribution of the obersvational network considered in France also has a clear effect on the interpolated value over the Pyrennes and the northeast of the Iberian Peninsula.

On the other hand, considering the ensemble version of E-OBS, E-OBS v17e, an experimental framework to evaluate the observational uncertainty has been defined, analizing if Iberia01 falls inside the ensemble given by E-OBS v17 and, then, if it could be considered a realization of the ensemble. In this case, we conclude that both datasets could be used indistinctly. First, note that the spread of the ensemble for precipitation has the same order of the mean value reflecting a large uncertainty for this variable, and questioning in this particular case the practical utility of this measure of uncertainty, in contrast to the one obtained for temperatures. In the case of precipitation (Figure 3, first row), the percentage of outliers considering only the wet-days ranges between $5\%$ and $25\%$ along the Peninsula. For temperatures (Figure 3, second to fourth rows), most of the area shows percentages less than $5\%$ of outliers, with only some regions previously identified (Guadalquivir basin, Pyrenees, etc.) presenting values larger than $50\% - 60\%$. 
[revised manuscript text omitted]

---

## Author Comment (AC2) · 16 Aug 2019

**Minor errors or clarifications:**

*p.2,l.5, "higher longitudinal and latitudinal resolution", I think?*

**Response:** We have changed "longitudinal" by "spatial".

*p.2,l.14, should be "has been analyzed"*

**Response:** We have modified the sentence accordingly.

*Figure 1 caption, "ised" should be "used"*

**Response:** We have modified the sentence accordingly.

*Table 1, RV50Yt - shouldn't this be the maximum daily 2-m air temperature?*

**Response:** To obtain the 50-years return values we consider the annual maximum of the corresponding variable, in our case daily precipitation and 2-meters daily mean temperature as is reflected in Table 1. The annual

maximum of 2-meters daily maximum temperature can be also considered but we have not analyzed this variable in the paper.

*p.6,l.8,"southwest to the northeast"*

**Response:** *We have modified the sentence accordingly.*

*p.6,l.14, "with" the main differences being...*

**Response:** *We have modified the sentence accordingly.*

*Discussion of Figure 2. It is hard to discern the differences. Difference maps would help to illustrate the main differences of interest between the datasets.*

**Response:** *We partially agree with the referee but we think that including a new table summarizing the differences between the different spatial patterns could have more added value than a new figure. So, we have added the following table to the manuscript:*

| Iberia01 | tas | RV50Yt | pr | RV50Yp | RR1 |
|---|---|---|---|---|---|
| MAE | 0.5404 | 1.6162 | 0.2888 | 20.1838 | 11.2783 |
| BIAS | -0.2099 | -1.3145 | 0.0881 | -17.8175 | 11.2763 |
| RMSE | 0.8902 | 2.9837 | 0.5651 | 28.1075 | 13.6863 |
| Correlation | 0.9422 | 0.7319 | 0.8496 | 0.8623 | 0.4304 |
| E-OBS v17 | tas | RV50Yt | pr | RV50Yp | RR1 |
| MAE | 0.8212 | 2.6219 | 0.4288 | 42.4205 | 10.0718 |
| BIAS | -0.2603 | -2.1310 | -0.2703 | -42.2184 | 9.9931 |
| RMSE | 1.1530 | 3.9377 | 0.7510 | 54.2519 | 12.5221 |
| Correlation | 0.8931 | 0.5403 | 0.7508 | 0.5891 | 0.4297 |
| E-OBS v17e | tas | RV50Yt | pr | RV50Yp | RR1 |
| MAE | 0.8260 | 2.6720 | 0.4357 | 46.9555 | 11.4778 |
| BIAS | -0.3341 | -2.3033 | -0.3021 | -46.7663 | 11.4641 |
| RMSE | 1.1811 | 4.0530 | 0.7600 | 58.0514 | 13.7543 |
| Correlation | 0.9047 | 0.5365 | 0.7560 | 0.5659 | 0.4374 |

**Table 1:** *Comparison between the spatial pattern of the different gridded datasets against the observations for the indices considered.*

*p.6,l.24, "all datasets show a clear overestimation" - why do you think this is? I don't understand why this would be for wet-day frequency, as it seems that this should come in a straightforward way from the dataset. How does interpolation or modelling introduce too many wet-days?*

**Response:** *Note that each grid point is obtained, in some way, as the spatial average of the surrounding stations. As a result, for each day if it has rained in one of the surrounding stations the interpolated value would be low but large enough to be considered as wet-day. A new table has been included in the new version of the manuscript to better illustrate this comment.*

*It would be interesting to see mean precipitation here as well, for each dataset.*

**Response:** *I am not sure what are the referee referring to as this index is included in Figure 2.*

*p.9, conclusions - the authors frequently refer to Iberia02, but that is the next paper isn't it? Up to here and in the title this is presenting Iberia01.*

**Response:** *We have corrected this error. In a first version, we decided to use Iberia02 to keep the coherence with the existing datasets, PT02 and Spain02, but we finally decided to use Iberia01.*

*p.9, ll.22-23 - I must misunderstand wet-days. I don't understand how the dry-days could be equivalent between datasets but the wet-days differ; I would have thought that wd = 365 - dd. This is likely just my deficiency, but others might also be confused here so some explanation would be good*

[Figure]

**Response:** *This comment was derived from the figure above that shows the percentage of dry/wet days well identified by E-OBS. In particular, the first row shows the quotient between the number of wet-days given by both Iberia01 and E-OBS (VP) and the number of wet-days of Iberia01 (P), and the second row the same information but for the dry-days. In this sense, the sum of both quantities should give the 100% of data. We have rewritten the sentence to avoid any misunderstanding.*

*p.6,l.21, double negative - I think it should be "either" and "or"*

**Response:** *We have modified the sentence accordingly.*

**Iberia01: A new gridded dataset of daily precipitation and temperatures over Iberia**

Sixto Herrera[a], Rita M. Cardoso[b], Pedro M.M. Soares[b], Fátima Espírito–Santo[c], Pedro Viterbo[c], and José M. Gutiérrez[d]

[a]Meteorology Group. Dept. of Applied Mathematics and Computer Science. Universidad de Cantabria. Santander, Spain
[b]Instituto Dom Luiz (IDL), Facultade de Ciências, Universidade de Lisboa, Lisboa, Portugal
[c]Instituto Português do Mar e da Atmosfera (IPMA), Lisboa, Portugal
[d]Meteorology Group. Instituto de Física de Cantabria, CSIC-University of Cantabria, Santander, Spain

**Correspondence:** Sixto Herrera (sixto.herrera@unican.es)

**Abstract.** The present work introduces a new observational gridded dataset produced using a dense network (thousands) of stations over the Iberian Peninsula (referred to as Iberia01, Gutiérrez et al. (2019), DOI: http://dx.doi.org/10.20350/digitalCSIC/ 8641), providing daily precipitation and temperatures for the period 1971-2015 at $0.1°$ regular (and $0.11°$ rotated CORDEX compliant) resolutions. A comparison with both the standard and ensemble version of the E-OBS v17 dataset (at $0.25°$ and $0.1°$ resolutions, respectively) is undertaken in order to assess observational uncertainty in this region. First, a standard comparison is performed for several weather indices, obtaining the differences between both datasets. Secondly, a new probabilistic intercomparison analysis is introduced, using the E-OBS ensemble (v17e) to characterize observational uncertainty and testing the hypothesis that Iberia01 is a realization of the ensemble (i.e. it falls within the observational uncertainty range provided by E-OBS). Finally, the possibility to increase the resolution of the dataset using the same interpolation approach is analyzed considering an extreme event of convective precipitation affecting the Iberian Peninsula and the auxiliary very high resolution grid ($0.01°$), built during the interpolation process to obtain the area-average representativity of the final dataset.

We show that Iberia01 produces more realistic patterns than E-OBS v17 in the case of precipitation for all the indices considered, although both are comparable for temperatures. These differences were assessed using a probabilistic approach based on the E-OBS ensemble. For precipitation, significant differences —at a 10% level— between both datasets were found for less than 25% of days over the Iberian Peninsula. For temperature, a very inhomogeneous pattern was obtained, with either a small (in most of the regions) or large fraction of significantly different days. The great uncertainty of the precipitation given by E-OBS ensemble, in which the standard deviation of the ensemble has the same order than the mean value, increases the significance of the results obtained for this variable reflecting the differences between both datasets.

KEY WORDS: *Observational uncertainty; E-OBS; ensemble; gridded observations; kriging; thin plate splines; extremes; Clime; Precipitation; Temperature*

*Copyright statement.* The Iberia01 gridded dataset is made available under the Open Database License. Any rights in individual contents of the database are licensed under the Database Contents License.

[revised manuscript text omitted]

standard deviations of the ensemble values) is of similar magnitude to the mean value (also with kernels of small uncertainty corresponding to stations) reflecting a large uncertainty for this variable.

Figure 3 shows the percentage of significantly different days for each gridbox, variable and season. For precipitation (first row), only Iberia01 wet-days were used in order to minimize the effect of the different wet-day frequencies. The differences for this variable exhibit a

5 homogeneous spatial pattern over the Peninsula with values around 10% in general; this is due to the large uncertainty of the daily E-OBS ensemble spread (see Fig. 2). Regarding the temperatures, most of the spatial pattern presents values close to zero, reflecting the similarity between both datasets for these variables. However, some local differences are found particularly for the mean (second row) and maximum (third row) temperatures, with the greatest values reached in the Pyrenean and Central ranges and the south coast of the Iberian Peninsula, in agreement with the differences shown in Figure 2. In this case, the ensemble uncertainty is in agreement with the differences between these

10 two datasets found in Figure 2.

To explore the possibility to increase the resolution of the Iberia01 grid, we consider the extreme event occurred the 4-5 November 1997 and compare the resulting values of the $0.1°$ grid with a higher resolution $0.03°$ one developed using the auxiliary $0.01°$ grid generated in the interpolation process. Tables 3 and 4, Figure 4 show the results obtained for the extreme event indicating that an increment of the Iberia01 resolution beyond 10 km has no clear impact in the effective resolution of the precipitation pattern. In particular, in spite of the clear

15 improvement of both versions of Iberia01 w.r.t. E-OBS v17e for all the parameters considered (see Table 3), there are only slight differences between both versions of Iberia01 when compared with observations.

Moreover, as it is reflected in Table 4, the spatial correlation between both resolutions is greater than 0.98 for both datasets, and any significant — at 5 % level — difference is found for the mean and variance of the spatial pattern according to the applied hypothesis test for two independent samples, the Student's t-test and Snedecor's F test for the mean and the variance, respectively.

[Figure]

**Figure 3.** Percentage of significantly different days between Iberia01 and E-OBS v17e for each gridbox, variable and season, defined as the Iberia01 daily values outside the the $P5 - P95$ percentile interval of the normal distribution given by the E-OBS ensemble, in the period 1970-2015 for wet-days (first row), and mean (second row), maximum (third row) and (fourth row) minimum temperatures.

[Figure]

**Figure 4.** Daily precipitation of the 4-5 November 1997 observed and given by E-OBS v17e, and a 3 km and 10 km version of Iberia01.

Note that the interpolation method, independently of the target resolution, is calibrated to reproduce the spatial dependence of the mean field of the target variable, which is usually greater than the grid resolution ($1°$ approximately in this case). Therefore, the effective resolution of purely interpolated gridded products is limited by this spatial value, which define the size of the kernels used for the interpolation process. As a result, in order to properly evaluate the convecting permitting CORDEX simulations, other approaches like regional reanalysis (e.g.

**Table 3.** Comparison between the spatial pattern of the different gridded datasets against the observations.

| Measure | Iberia01 3 km | Iberia01 10 km | E-OBS v17 (25 km) | E-OBS v17e (10 km) |
|---------|--------------|----------------|-------------------|---------------------|
| MAE | 3.3861 | 4.8142 | 11.0998 | 10.9936 |
| BIAS | 0.2352 | 0.7046 | -1.7382 | -1.4243 |
| RMSE | 6.5430 | 8.9360 | 19.0300 | 18.7614 |
| CORR | 0.9746 | 0.9522 | 0.7625 | 0.7691 |

Häggmark et al., 2000) or methods combining interpolation and analysis as the proposed by Quintana-Seguí et al. (2017) and Peral et al. (2017), among others, should be used.

**Table 4.** Comparison between the spatial pattern of the different resolutions considered to analyse the effective resolution of the gridded datasets.

| Measure | Iberia01 3 km vs. Iberia01 10 km | E-OBS v17 (25 km) vs. E-OBS v17e (10 km) |
|---------|----------------------------------|------------------------------------------|
| CORR | 0.9855 | 0.9912 |
| t test (H) | 0 | 0 |
| F test (H) | 0 | 0 |

**4 Conclusions and Discussion**

In this work a new gridded dataset for the Iberian Peninsula and the Balearic Islands based on a quality-controlled and dense station network
5 has been described and compared with E-OBS v17, considering both the standard and the ensemble version of this product, to reflect and analyze the observational uncertainty related with both datasets.

On the one hand, Iberia01 is able to reproduce the spatial pattern and intensity of both the mean and extreme regimes of precipitation and temperature, in terms of the weather indices defined in Table 1, including extreme events as the one occurred the 4-5 November 1997 shown in the Figure 4. For the weather indices considered, E-OBS v17 tends to underestimate the extremes and soften the spatial pattern of
10 precipitation, in agreement with other previous studies (Herrera et al., 2012). It is however more similar to Iberia01 in the case of temperature indices, with the main differences appearing in the Guadalquivir and Guadiana basins, and the Pyrenean range. In addition, both datasets present large differences for wet-days (see Figure 2), with E-OBS v17 identifying less than the 70% of the observed wet-days all around the Peninsula and falling up to the 40% in Summer. Note that the complex orography and the influence of both the Atlantic Ocean and the Mediterranean Sea modulate the precipitation over the Iberian Peninsula, leading to particular regimes, as the cold drop in the east coast,
15 that a continental adjustment of the interpolation model is not able to reproduce, even more when a low-dense observational network is considered. In this sense, the large increase of rain gauges considered in Iberia01, when compared with E-OBS, give rise to a much improved precipitation rendering. In the case of temperature, although the observational network considered is similar in both cases, the pattern tends to be more orographic in E-OBS v17 due to the continental adjusment of the interpolation method that overrates this component avoiding

regional behaviors. In addition, the contribution of the obersvational network considered in France also has a clear effect on the interpolated value over the Pyrennes and the northeast of the Iberian Peninsula.

On the other hand, considering the ensemble version of E-OBS, E-OBS v17e, an experimental framework to evaluate the observational uncertainty has been defined, analizing if Iberia01 falls inside the ensemble given by E-OBS v17 and, then, if it could be considered a realization of the ensemble. In this case, we conclude that both datasets could be used indistinctly. First, note that the spread of the ensemble for precipitation has the same order of the mean value reflecting a large uncertainty for this variable, and questioning in this particular case the practical utility of this measure of uncertainty, in contrast to the one obtained for temperatures. In the case of precipitation (Figure 3, first row), the percentage of outliers considering only the wet-days ranges between $5\%$ and $25\%$ along the Peninsula. For temperatures (Figure 3, second to fourth rows), most of the area shows percentages less than $5\%$ of outliers, with only some regions previously identified (Guadalquivir basin, Pyrenees, etc.) presenting values larger than $50\% - 60\%$. 
[revised manuscript text omitted]

---

## Author Comment (AC4) · 16 Aug 2019

Dear Referee, Many thanks for the guidelines and constructive comments to our manuscript. Please find attached a revised version of the manuscript and the point-by-point responses to your comments We appreciate your work in helping us to improve the manuscript.

We hope the revised manuscript is now acceptable for publication in Earth System Science Data. All authors agree on the current form of the manuscript.

Dr. Sixto Herrera, on behalf of the authors.

---

## Author Response (AR1)

**Title: Iberia01: A new gridded dataset of daily precipitation and temperatures over Iberia**
**Author(s): Sixto Herrera et al.**
**MS No.: essd-2019-95**
**MS Type: Data description paper**
**Iteration: Revised Submission**

Dear Editor,

We are submitting a revised version of the above mentioned manuscript. We are confident that we have satisfactorily addressed all reviewers' comments and that the revised manuscript will meet the high quality standards of ESSD. Please find below the point-by-point responses to the reviewers' comments and the new manuscript with (and without) tracked changes. We hope the revised manuscript is now acceptable for publication in *Earth System Science Data*. All authors agree on the current form of the manuscript.

Dr. Sixto Herrera, on behalf of the authors.

**Anonymous Referee #1:**

*Review of "Iberia01: A new gridded dataset of daily precipitation and temperatures over Iberia"*
*This paper presents a gridded climatological data product for the Iberian peninsula called "Iberia01" which appears to be a revision of a previous data product called "Spain02", using the same station network and interpolation methods, except including "orography" (elevation?) as covariate in the thin-plate spline step. Iberia01 is compared against a standard (E-OBS), finding more or less similar predictions except in some specific locations for specific climate variables. For precipitation it is found that the higher-resolution Iberia01 predictions show more small-scale variation than the coarse E-OBS product, at least in the case of a specific major precipitation event. The construction and integrity of the dataset appear to be well-done overall, although much of the methods refer to a previously published paper by the same authors. The figures are well-done as well. As a presentation of a new dataset, I think this paper should suffice (with some major revisions, clarifications, etc), although it is not clear whether any of the techniques or analyses are particularly novel.*

*Interactive comment on **Earth Syst. Sci. Data Discuss., https://doi.org/10.5194/essd-2019-95, 2019.***

*specific lines referenced as (pg:line)*

**Response:** We thank the reviewer for the comments and the time devoted to our paper. Please, see below our point-by-point responses and the changes highlighted as tracked changes in the new version of the manuscript.

**Major Issues:**

***Reviewer's Comment:*** *None of the links (p11: 4-6) to the data worked for me. The first sent me to a generic landing page in Portuguese. The second sent me to a site where the data was embargoed and required a login account. The third sent an error message. I assume based on the R code that the third access point requires authentication with Santander and requests via a specialized R function. The AEMET link (pg 10) sent me to another landing page where it was unclear how to find the dataset. Either way, I could not access the data.*

**Response:** We thank the referee for pointing out this comment. In the original manuscript we included several alternative potential access points for the dataset (corresponding to the webs and services of the different institutions involved in the work). In the revised version we leave just one of them, the DIGITAL.CSIC service for open science, which was embargoed at the time of revision of the original manuscript. The embargo was established to prevent the use of the dataset before the publication of this reference paper. The embargo has now expired and the data is now freely available at the provided link. Moreover, we have included also some metadata information on the stations used which is also available at the link.

Alternatively, a service for remote access is provided via the Santander Climate Data Service (CDS), which requires a (free online) registration (a link to a page with instructions is provided in the paper). The section now reads:

> *"All the datasets used in this work are publicly available. On the one hand, the Iberia01 dataset is publicly available through the DIGITAL.CSIC open science service (Herrera et al., 2019, DOI: http://dx.doi.org/10.20350/digitalCSIC/8641). Moreover, a THREDDS remote access to this dataset is available from the Santander Climate Data Service, via the User Data Gateway (instructions at http://meteo.unican.es/udg-wiki). On the other hand, the E-OBS v17 dataset is remotely available through the KNMI's THREDDS server http://opendap.knmi.nl/knmi/thredds/e-obs/e-obs-catalog.html and the ensemble version E-OBS v17e is available through the Copernicus' Climate Change Service http://surfobs. climate.copernicus.eu.*

*The R code needed to partially reproduce the results of this paper (for the remotely accessible datasets Iberia01 and E-OBS v17) is publicly available at https://github.com/SantanderMetGroup/notebooks, building on the remote data services above described and on the climate4R R framework (Iturbide et al., 2019)."*

**Reviewer's Comment:** *The authors state that the E-OBS dataset is taken as a benchmark (4:7) but later claim that the dataset is biased for key variables (6:22). It is not clear in the methods how this assessment is made or quantified.*

**Response:** The E-OBS dataset was used as benchmark in the manuscript because it is considered the reference dataset at European scale in many studies (it has more than 1490 citations in Scopus as by August 2019). However, as discussed in the introduction of the paper, at national or regional scale E-OBS presents some known biases, mainly in regions with complex orography and/or with low stations' density. Therefore, besides the comparison plots shown in Figure 2, we wanted to include some quantitative assessment of the differences. However, we agree with the referee that that no information is provided on how the assessments on 6:22-24 were obtained:

*"E-OBS underestimates mean precipitation by 15 - 20% (mean relative bias, for E-OBSv17 - v17e, respectively), particularly in the Central System range of the Iberian Peninsula, and 50-year return values by 42 - 47% (mean relative bias),"*

These values were directly calculated from the spatial mean values of the different indices shown in the different panels in Figure 2 (e.g. a mean value of 1.6mm for E-OBS v17 and 1.9mm for the Stations, resulting in ~15% relative difference). However, we agree that in the original manuscript these results were difficult to follow.

In the revised version we have clarified this, extending the analysis to take into account the different nature of the gridded (Iberia01 and E-OBS) and point-based (station) datasets shown in Figure 2. Therefore, besides calculating (and showing in the different panels) the spatial mean values of the gridded datasets, we included a second value comparable with the result for the stations, averaging over the number of stations considering the value of nearest gridbox to the local station. This provides a fair estimate of the biases of the gridded products, when compared with the original station-based one. A table with different statistics computed following this approach was included in the responses to the interactive reviewers' comments (see below). However, in the revised manuscript we have decided to keep the analysis as simple as possible and included only the spatial mean values (both over the whole grid, or over the stations' corresponding gridboxes) in the different panels of Figure 2, together with a proper explanation of these numbers included in the figure caption:

*"Climatology (mean) of the different temperature (top) and precipitation (bottom) indices defined in Table 1, from the stations (local values), Iberia01 (0.1° resolution), E-OBS v17 (0.2°) and E-OBS v17e (0.1°), in rows. For the ensemble E-OBS version (v17e), the climatology of daily standard deviations of the ensemble values is also shown (characterizing E-OBS observational uncertainty). The numbers shown in each panel correspond to the spatial mean values, calculated for gridded datasets averaging over all gridboxes (top, in italics) or over the gridboxes nearest to the stations (bottom), to provide a fair comparison with the station mean values in the top panels (in particular these numbers are used to assess the biases of the different gridded datasets)."*

The procedure followed to assess the different biases (based on Figure 2 information) are now clearly described in the revised manuscript.

| Iberia01 | tas | RV50Yt | pr | RV50Yp | RR1 |
|---|---|---|---|---|---|
| MAE | 0.5404 | 1.6162 | 0.2888 | 20.1838 | 11.2783 |
| BIAS | -0.2099 | -1.3145 | 0.0881 | -17.8175 | 11.2763 |
| RMSE | 0.8902 | 2.9837 | 0.5651 | 28.1075 | 13.6863 |
| Correlation | 0.9422 | 0.7319 | 0.8496 | 0.8623 | 0.4304 |
| **E-OBS v17** | **tas** | **RV50Yt** | **pr** | **RV50Yp** | **RR1** |
| MAE | 0.8212 | 2.6219 | 0.4288 | 42.4205 | 10.0718 |
| BIAS | -0.2603 | -2.1310 | -0.2703 | -42.2184 | 9.9931 |
| RMSE | 1.1530 | 3.9377 | 0.7510 | 54.2519 | 12.5221 |
| Correlation | 0.8931 | 0.5403 | 0.7508 | 0.5891 | 0.4297 |
| **E-OBS v17e** | **tas** | **RV50Yt** | **pr** | **RV50Yp** | **RR1** |
| MAE | 0.8260 | 2.6720 | 0.4357 | 46.9555 | 11.4778 |
| BIAS | -0.3341 | -2.3033 | -0.3021 | -46.7663 | 11.4641 |
| RMSE | 1.1811 | 4.0530 | 0.7600 | 58.0514 | 13.7543 |
| Correlation | 0.9047 | 0.5365 | 0.7560 | 0.5659 | 0.4374 |

*Table 1: Comparison between the spatial pattern of the different gridded datasets against the observations for the indices considered [not included in the revised manuscript, only some of the results].*

**Reviewer's Comment:** *It would be nice to provide some ideas explaining the specific deviations (e.g. along the coast) between Iberia01 and E-OBS in the discussion.*

**Response:** We have included in the conclusions and discussion section a paragraph discussing this point and pointing out to possible reasons for the observed differences between both datasets:

"Note that the complex orography and the influence of both the Atlantic Ocean and the Mediterranean Sea modulate the precipitation over the Iberian Peninsula, leading to particular regimes, as the cold drop in the east coast, that a continental adjustment of the interpolation model is not able to reproduce, particularly when a low-dense observational network is considered. In this sense, the large increase of rain gauges considered in Iberia01, when compared with E-OBS, give rise to a much improved precipitation rendering. In the case of temperature, although the observational network considered is similar in both cases, the pattern tends to be more orographic in E-OBS v17 due to the continental adjustment of the interpolation method that overrates this component avoiding regional behaviors. In addition, the contribution of the observational network considered in France also has a clear effect on the interpolated value over the Pyrenees and the northeast of the Iberian Peninsula."

**Reviewer's Comment:** *The assessment of resolution for the convective rain event is unsatisfying. First, why do the authors present the 20 km product (v17) and not the 10 km product (v17e) for E-OBS, which seems like a much better comparison?*

**Response:** *Following the referee's comment we have updated the figure including the high-resolution version of E-OBS (~10 km) but, as can be seen in the following figure, the conclusions have not changed.*

[Figure]

*Figure: Comparison between both resolutions of the E-OBS v17e dataset.*

**Reviewer's Comment:** *Second, the authors claim that the difference in resolution for Iberia01 10 vs 3 km resolution does not matter, but this is not quantitatively examined or explained in any way. Are the authors using gestalt, I assume?*

**Response:** *Although some further analysis was included in the responses of the interactive comments, we found this whole analysis unsatisfying (in agreement with the referee) and we have downgraded this topic in the revised manuscript to a simple illustrative example. Therefore, we have removed Sec. 2.5 (effective resolution) and included this discussion in the section describing the gridding method (and the different resolutions of the intermediate and final products). The objective of this example is to provide a simple example illustrating graphically the effective resolution of the different products.*

**Reviewer's Comment:** *There are numerous grammatical errors, run-on sentences and awkward phrasings throughout. I have noted some below, but not all. The writing is good in terms of logic, but needs a careful proofread (possibly by a native English speaker) before it is publishable.*

**Response:** We have revised the text carefully to eliminate grammatical errors and to rephrase awkward sentences.

**Reviewer's Comment:** *The methods frequently refer to Herrera 2011, 2012. However, more brief descriptions of these methods would be helpful, such as the QC protocols.*

**Response:** We have extended the description of the quality control procedure in the new version of the manuscript. In addition, two files with metadata information for each of the stations (geospatial information, start/end year, missing data for the whole period, number of years with less than 10% of missing data) have been included in the dataset one for precipitation and another for temperature. These files are available from the same DOI http://dx.doi.org/10.20350/digitalCSIC/8641 and provide detailed information on the characteristics of the observational networks used to build the gridded dataset.

> *"To keep consistency with previous datasets, the final network was obtained applying the same quality control used to build Spain02 (see Herrera, 2011, Herrera et al. 2012, for a detailed description), which requires stations with at least 15 (40) years in the period 1951-2015 with less than 10% yearly missing precipitation (temperature) data. The resulting observational network includes 3486 and 275 stations for precipitation and temperature, respectively, as shown in Figure 1(a-b). Note that detailed metadata for each station, including geographical and data availability information, is provided as part of the dataset in the same repository."*

**Minor Issues:**

*Where did the 'orography' dataset come from? Isn't this just elevation?*

**Response:** We have just considered elevation, given by the Global Digital Elevation Model (GTOPO30) which provides gridded 30 arc seconds (~1 km) elevation worldwide. We have changed 'orography' by elevation accordingly in several parts of the manuscript.

*The R code (pg 11, line 10) Seems to only calculate and visualize climatologies but doesn't actually do any direct comparisons.*

**Response:** We have rewritten and simplified the notebook provided with the paper. Now it computes the differences of the datasets and allows to easily calculate further indices to extend the analysis provided in the paper. We have included a comment on this in the "code and data availabililty" section:

> *"The R code needed to partially reproduce the results of this paper (for the remotely accessible datasets Iberia01 and E-OBS v17) is publicly available at https://github.com/SantanderMetGroup/notebooks, building on the remote data services above described and on the climate4R R framework (Iturbide et al., 2019)."*

*(6:10) How exactly?*

**Response:** *We have downgraded the topic of effective resolution in the revised manuscript to a simple illustrative example (see previous comments).*

*paragraph (6:29ff) move to methods*

**Response:** We have modified the manuscript accordingly and included this paragraph at the end of the section "E-OBS Gridded Datasets (v17 and v17e)".

*(6:30): "can be considered a realization of" seems like an awkward way to phrase it. Why not 'differs significantly from'?*

**Response:** We have modified the sentence accordingly.

*(7:2) "thus questioning" – Move interpretations like this to the discussion and flesh them out. I would tend to disagree with this statement as stands.*

**Response:** *We have modified the sentence accordingly.*

**Technical issues:**

*Please remove "In order" from all sentences beginning with "In order to", as this is redundant.*

**Response:** *We have modified the manuscript accordingly.*

*(1:9) Run on sentence*

**Response:** *We have rewritten the sentence.*

*(1:11) omit "As a result"*

**Response:** *We have modified the sentence accordingly.*

*(1:15) rephrase*

*Response: The paragraph has been reformulated.*

*(2:6) expands -> includes*

**Response:** *We have modified the sentence accordingly.*

*(2:11-12) reference for this assertion?*

**Response:** We have included the citation's number of E-OBS in Scopus (1491 citations). In addition, we have modified the sentence to better clarify its meaning:
> *"With more than 1490 citations in Scopus (as by August 2019), this is the most used climate reference for European climate studies."*

*(2:14) omit 'the'*

**Response:** We have modified the sentence accordingly.

*(2:19) 'smooths' awkward term here*

**Response:** We have rewritten the sentence accordingly: "..., a reduction in the density of stations decreases the variability of both precipitation and temperature with large implications in the representation of extremes."

*(3:6) include 'and' between citations*

**Response:** We have modified the sentence accordingly.

*(3:15) Run on sentences*

**Response:** We have modified the sentence.

*(3:20) Is this really the first? Seems like there are others, referenced in the same sentence (PT02)*

**Response:** *The Iberia01 dataset is the first gridded dataset built ad hoc for the Iberian Peninsula. The previous IB02 was a dataset of opportunity created by joining two datasets (PT02 and Spain02). We have clarified this in the introduction:*
> "...for continental Portugal using more than 400 stations (PT02). Both datasets had consistent grids (with 0.2º resolution) and time periods (1950-2003) and were combined to build a gridded precipitation dataset of opportunity for the Iberian peninsula (IB02). However, this is not an homogeneous product for the Iberian peninsula due to the discrepancies existing between the two datasets near the borders, particularly in the northern mountain"

*(4:8) Replace 'first dataset', 'this one' etc with specific title of each. Confusing*

**Response:** We have modified the sentence accordingly.

*(5:10) "temperature was built"*

**Response:** We have modified the sentence accordingly.

*(6:6) Run on sentence*

*Response: We have modified the sentence accordingly.*

*(6:13) How were clims aggregated?*

**Response:** The climatologies have been obtained averaging the annual values of the indices (tas, pr and RR1). In the case of the 50-years return value the index is representative of all the period, so it is its own climatological value. This has been clarified in the caption of table 2.

*(6:14) "the main differences being"*

*Response: We have modified the sentence accordingly.*

*(6:29) "we used"*

**Response:** We have modified the sentence accordingly.

*(9:7) on -> of*

**Response:** We have modified the sentence accordingly.

*(9:28) and (9:30) – these sentences are both difficult to understand.*

**Response:** The conclusions have been substantially modified in the revised version.

**Anonymous Referee #2:**

*Review of "Iberia01: A new gridded dataset of daily precipitation and temperatures over Iberia"*

*The authors present an extensive, long-term dataset of temperature and precipitation in Iberia, based on a combination and extension of datasets from Spain and Portugal. While it is not dramatically new from previous data compilations by the senior author and his colleagues, they do introduce higher resolution and some new analysis. For instance, having elevation as a covariate in the interpolation procedure is a valuable improvement.*

*While incremental, this is a valuable dataset that can be used in a wide range of applications. I don't know of a network of observations this extensive, dense, and long-running anywhere in the world. While it is a shame to see the number of observations degrade in recent years, this is a valuable dataset that can be used for either weather or climate analyses. I can certainly see the value of this dataset as a test of the CORDEX high-resolution simulations. The paper is well-written: clear and concise. I recommend publication with minor revisions.*

*Interactive comment on Earth Syst. Sci.* **Data Discuss.,** **https://doi.org/10.5194/essd-2019-95****, 2019.**

**Response:** We thank the reviewer for the comments and the time devoted to our paper. Please, see below our point-by-point responses and the changes highlighted as tracked changes in the new version of the manuscript.

**Minor errors or clarifications:**

*p.2,l.5, "higher longitudinal and latitudinal resolution", I think?*

**Response:** We have changed "longitudinal" by "spatial".

*p.2,l.14, should be "has been analyzed"*

**Response:** We have modified the sentence accordingly.

*Figure 1 caption, "ised" should be "used"*

**Response:** We have modified the sentence accordingly.

*Table 1, RV50Yt - shouldn't this be the maximum daily 2-m air temperature?*

**Response:** To obtain the 50-years return values we consider the annual maximum of the corresponding variable, in our case daily precipitation and 2-meters daily mean temperature as is reflected in Table 1. The annual maximum of 2-meters daily maximum temperature can be also considered but we have not analyzed this variable in the paper.

*p.6,l.8,"southwest to the northeast"*

*Response: We have modified the sentence accordingly.*

*p.6,l.14, "with" the main differences being...*

*Response: We have modified the sentence accordingly.*

*Discussion of Figure 2. It is hard to discern the differences. Difference maps would help to illustrate the main differences of interest between the datasets.*

***Response:*** Calculating and showing differences for maps of different nature (stations vs. gridded) and resolutions would involve regridding and would make difficult to interpret the obtained results. Therefore, we have opted to keep the same maps and 1) provide further intercomparisson analysis as detailed below and 2) provide the code required for reproducibility in a user-friendly notebook (described in the "code and data availabililty" section) which allows to easily compute the differences, giving flexibility for the different regridding options (when needed). Now that the notebook computes the differences of the datasets and allows to easily calculate further indices to extend the analysis provided in the paper. The figure below is included in the notebook and shows the differences between Iberia01 and E-OBS v17 (regridded to the Iberia01 grid).

**Bias (1971–2010)**

[Figure]

Regarding the additional intercomparisson analysis, besides the spatial means of the different products shown in the original manuscript, we have extended the analysis to take into account the different nature of the gridded (Iberia01 and E-OBS) and point-based (station) datasets shown in Figure 2. Therefore, for the gridded datasets, we include now a second value comparable with the result for the stations, averaging over the number of stations considering the value of nearest gridbox to the local station. This provides a fair estimate of the biases of the gridded products, when compared with the original station-based one. A table with different statistics computed following this approach was included in the responses to the interactive reviewers' comments (see below). However, in the revised manuscript we have decided to keep the analysis as simple as possible and included only the spatial mean values (both over the whole grid, or over the stations' corresponding gridboxes) in the different panels of Figure 2, together with a proper explanation of these numbers included in the figure caption.

| Iberia01 | tas | RV50Yt | pr | RV50Yp | RR1 |
|---|---|---|---|---|---|
| MAE | 0.5404 | 1.6162 | 0.2888 | 20.1838 | 11.2783 |
| BIAS | -0.2099 | -1.3145 | 0.0881 | -17.8175 | 11.2763 |
| RMSE | 0.8902 | 2.9837 | 0.5651 | 28.1075 | 13.6863 |

| Iberia01 | tas | RV50Yt | pr | RV50Yp | RR1 |
|---|---|---|---|---|---|
| Correlation | 0.9422 | 0.7319 | 0.8496 | 0.8623 | 0.4304 |
| **E-OBS v17** | **tas** | **RV50Yt** | **pr** | **RV50Yp** | **RR1** |
| MAE | 0.8212 | 2.6219 | 0.4288 | 42.4205 | 10.0718 |
| BIAS | -0.2603 | -2.1310 | -0.2703 | -42.2184 | 9.9931 |
| RMSE | 1.1530 | 3.9377 | 0.7510 | 54.2519 | 12.5221 |
| Correlation | 0.8931 | 0.5403 | 0.7508 | 0.5891 | 0.4297 |
| **E-OBS v17e** | **tas** | **RV50Yt** | **pr** | **RV50Yp** | **RR1** |
| MAE | 0.8260 | 2.6720 | 0.4357 | 46.9555 | 11.4778 |
| BIAS | -0.3341 | -2.3033 | -0.3021 | -46.7663 | 11.4641 |
| RMSE | 1.1811 | 4.0530 | 0.7600 | 58.0514 | 13.7543 |
| Correlation | 0.9047 | 0.5365 | 0.7560 | 0.5659 | 0.4374 |

*Table 1: Comparison between the spatial pattern of the different gridded datasets against the observations for the indices considered [not included in the revised manuscript, only some of the results].*

*p.6,l.24, "all datasets show a clear overestimation" - why do you think this is? I don't understand why this would be for wet-day frequency, as it seems that this should come in a straightforward way from the dataset. How does interpolation or modelling introduce too many wet-days?*

**Response:** Note that each grid point is obtained, in some way, as the spatial average of the surrounding stations. As a result, for each day if it has rained in one of the surrounding stations the interpolated value would be low but large enough to be considered as wet-day.

*It would be interesting to see mean precipitation here as well, for each dataset.*

**Response:** We are not sure what the referee is referring to as this index is included in Figure 2.

*p.9, conclusions - the authors frequently refer to Iberia02, but that is the next paper isn't it? Up to here and in the title this is presenting Iberia01.*

**Response:** We have corrected this error. In a first version, we decided to use Iberia02 to keep the coherence with the existing datasets, PT02 and Spain02, but we finally decided to use Iberia01 to make emphasis on the higher resolution.

*p.9, ll.22-23 - I must misunderstand wet-days. I don't understand how the dry-days could be equivalent between datasets but the wet-days differ; I would have thought that wd = 365 - dd. This is likely just my deficiency, but others might also be confused here so some explanation would be good*

**Response:** This comment was misleading since it referred to the coincidence (hit rates) of dry and wet days in the two datasets (E-OBS and Iberia01). This comment was based on the results from the figure below (not included in the paper) that shows the percentage of dry/wet days well identified by E-OBS, considering Iberia01 as the reference. In particular, the first row shows the ratio between the number of wet-days coincident in both Iberia01 and E-OBS (VP) and the number of wet-days of Iberia01 (P), and the second row the same information but for the dry-days. In this sense, the sum of both quantities should give the 100% of data. The whole conclusions section has been rewritten and simplified and the above discussion has been removed.

[Figure]

*p.6,l.21, double negative - I think it should be "either" and "or"*

*Response: We have modified the sentence accordingly.*

**Iberia01: A new gridded dataset of daily precipitation and temperatures over Iberia**

Sixto Herrera[a], Rita M. Cardoso[b], Pedro M.M. Soares[b], Fátima Espírito–Santo[c], Pedro Viterbo[c], and José M. Gutiérrez[d]

[a]Meteorology Group. Dept. of Applied Mathematics and Computer Science. Universidad de Cantabria. Santander, Spain
[b]Instituto Dom Luiz (IDL), Facultade de Ciências, Universidade de Lisboa, Lisboa, Portugal
[c]Instituto Português do Mar e da Atmosfera (IPMA), Lisboa, Portugal
[d]Meteorology Group. Instituto de Física de Cantabria, CSIC-University of Cantabria, Santander, Spain

**Correspondence:** Sixto Herrera (sixto.herrera@unican.es)

**Abstract.** The present work  presents a new observational gridded dataset (referred to as Iberia01) for daily precipitation and temperatures produced using a dense network (thousands) of stations over the Iberian Peninsula  for the period 1971-2015 at 0.1° regular (and 0.11°  CORDEX-compliant rotated) resolutions.  We analyze mean and extreme indices and compare the results with the E-OBS v17 dataset (using both the standard and ensemble versions, at 0.25° and 0.1° resolutions, respectively) , in order to assess observational uncertainty in this region.  We show that Iberia01 produces more realistic precipitation patterns than E-OBS for the mean and extreme indices considered, although both are comparable for temperatures. To assess the differences between  these datasets, a new probabilistic intercomparison analysis  was performed, using the E-OBS ensemble (v17e) to characterize observational uncertainty and testing  whether Iberia01  falls within the observational uncertainty range provided by E-OBS. In general, uncertainty values are large in all the territory, with the exception of a number of kernels where the uncertainty is small, corresponding to the stations used to build the

 grid. For precipitation, significant differences —at a 10% level—  between both datasets  were found for less than 25% of days over the Iberian Peninsula. For temperature, a very inhomogeneous  spatial pattern was obtained, with either a small (in most of the regions) or large fraction of significantly different days.

than the mean value, increases the significance of the results obtained for this variable reflecting the differences between both datasets. , thus indicating sensible regions for observational uncertainty.

Iberia01 is publicly available (Herrera et al., 2019a, DOI: http://dx.doi.org/10.20350/digitalCSIC/8641).

KEY WORDS: *Observational uncertainty; extremes; gridded observations; kriging; thin plate splines; E-OBS*

5    *Copyright statement.* The Iberia01 gridded dataset is made available under the Open Database License. Any rights in individual contents of the database are licensed under the Database Contents License.

[revised manuscript text omitted]